# Assessment of the Performance of FireFOAM in Simulating a Real-Scale Fire Scenario Using High Resolution Data

Wolfram Jahn [1,2,*,†], Rafael Zamorano [1,†,‡], Ignacio Calderón [1,2], Raimundo Claren [1] and Benjamín Molina [1]

1. Department of Mechanical and Metallurgical Engineering, Pontificia Universidad Católica de Chile, Santiago 7820436, Chile; razamorano@uc.cl (R.Z.); iacalderon@uc.cl (I.C.); rclaren@uc.cl (R.C.); benmolina@uc.cl (B.M.)
2. Centro Nacional de Excelencia Para la Madera (CENAMAD), Pontificia Universidad Católica de Chile, Santiago 7820436, Chile
* Correspondence: wjahn@uc.cl
† These authors contributed equally to this work.
‡ Current address: Colbun S.A., Av. Apoquindo 4775, Las Condes 7580097, Chile.

**Abstract:** An assessment of the performance of FireFOAM in simulating a large-scale compartment fire scenario is presented in this study, using the Edinburgh Tall Building Fire Test I (2017) as the basis for evaluation. Different mesh geometries and sizes are tested, and both theory-based and experiment-based validation approaches are employed. The theory-based validation revealed that the implemented finite volumes method is generally conservative, but unaccounted deviations of up to 20% for the heat release rate were observed due to errors in numerically modelling subgrid phenomena, particularly with tetrahedral meshes. In the experiment-based validation, the simulated data showed good agreement with experimental measurements for flow patterns inside the compartment, neutral plane height, and temperatures outside the ceiling jet. However, there were relatively large errors in incident radiation in the hot gas zone, thermal boundary layer transient temperatures, and compartment inflow/outflow velocities. Systematic errors were attributed to deficient heat transfer boundary conditions and under-estimated air entrainment. The study also identified ways to improve run-time efficiency by implementing parallel processing or reducing solid angles in FVDOM, although using large meshes (30 cm and 40 cm cell size) resulted in faster run-times at the cost of accuracy.

**Keywords:** full-scale compartment fire; LES; OpenFOAM; ETFT test 1; run-time

## 1. Introduction

Computational simulations of fire dynamics are an integral part of fire safety engineering, with applications both in design and in forensic analysis of fire accidents (e.g., [1,2]). Several authors have proposed to extend the use of fire simulations to the assistance of emergency response services, combining fire models with sensor data to speed up computations (e.g., [3–5]). The emergence of deep learning techniques for fire forecast open a new field of application for fire simulations, since the lack of large quantities of data to train the learning algorithms requires the use of simulations as an alternative [6,7].

There are several types of models available that can be used for simulating fire dynamics. While simplified two-zone models have short running times, they only yield very limited output (typically average hot layer temperature and smoke-layer height), resulting in a trend towards ever more sophisticated fire models based on computational fluid dynamics (CFD).

The different time and length scales involved in fire dynamics pose a special challenge to CFD-based numerical simulations. The computational domain for the simulation of a meaningful fire scenario typically stretches over several metres in each direction, and

the fire duration can extend well over 10 min. Many of the physical processes that govern fire dynamics, however, occur on a microscopic scale within a short period of time (e.g., presence of a flame sheet in a certain location). While several phenomena of interest, such as smoke movement or local temperatures, have been reproduced reasonably well by several authors using adequate modelling techniques, they usually rely on a detailed knowledge of the heat release rate (HRR) of the fire [8–10]. The prediction of the fire growth itself depends on the complex interaction between flame and fuel, and cannot generally be performed in full-scale fire scenarios [8,11], rendering purely predictive fire development modelling (i.e., without input parameter calibration) virtually impossible.

An extensive run-time is one of the limiting factors regarding computational fire simulations. While sufficient time could potentially be provisioned for design applications in fire protection, fire forecast simulations for the assistance of emergency response systems inherently depend on faster-than-real-time completion (it would not be a forecast if it arrived after the event took place). This is a specially restrictive condition in the context of data-driven approaches, where parameters are to be calibrated with live incoming data from the fire scenario. So far, achieving real-time fire scenario simulations with reasonable predictions is restricted to small-scale scenarios with the use of high performance computing facilities. In order to make progress in reducing the computing time for emergency response systems in real-scale scenarios, the assessment of the available solvers' limitations is crucial.

There are several *out of the box* CFD solvers available that can be used for fire simulations. Even though general purpose commercial solvers, such as Ansys CFX or Fluent, are not specifically designed for such simulations, some aspects of fire modelling, such as smoke propagation, can be implemented conveniently with suitable user-defined submodels (see, e.g., [12,13]). The most commonly used CFD package for fire simulations is probably NIST's Fire Dynamics Simulator (in its current version, FDS6.7) [14], specifically designed for this purpose and widely validated [15]. FDS6.7 performs fire simulations very efficiently, partly due to its relatively simple meshing philosophy of uniform rectangular cells (non-uniform meshing is, in principle, possible, but is elementary and thus has a limited range or applicability) FireFOAM, an OpenFOAM-based CFD solver designed for the simulations of turbulent diffusion flames, has the capacity to implement irregular meshes and adjust the mesh size during the simulation, which enables the application of dynamic mesh refinement (DMR). Similar techniques have been shown to produce important speed-ups of fire simulations (e.g., [16]), and are a feature worth exploring. In order to enable a more complex use of FireFOAM, it is necessary to assess its capabilities and performance.

The performance of FireFOAM has been studied before by comparing its output to experimental data [17–20], all of them were constrained by one of the following limitations:

- Only global fire parameters were tested, e.g., evolution of global heat release rate [20], therefore, there was a need for more aggressive evaluations of the LES-based capabilities [21].
- Validation was performed based on low spatial resolution measurements and/or literature correlations [22].
- Validation was performed for small–medium-scale fires, e.g., pool-fire configurations of less than 1 m in diameter [17].

The high-resolution multi-variable measurement data collected at the "Edinburgh Tall Building Fire Tests" (ETFT) compartment fire experiments, and published in [23] (data available on https://datashare.ed.ac.uk/handle/10283/3233, accessed on 16 June 2023) present an opportunity to validate FireFOAM without incurring the previously stated limitations. With multiple variables measured in high spatial resolution, these experiments present a challenging validation scenario for evaluating FireFOAM's capabilities. In large-eddy simulation (LES) applications for large-scale industrial fires, spatial scales vary significantly and, therefore, small cell sizes cannot be used throughout, so subgrid-scale modelling becomes more challenging and important.

A validation study of FireFOAM for large-scale open-plan compartments is presented here, based on the ETFT experimental measurements, particularly test I. ETFT test I consisted of a fully developed, fuel-controlled fire with complete and static ventilation, presenting a challenging but adequate scenario for the validation. Additionally, the performance of FireFOAM in terms of run-time, mesh convergence, and sensitivity to the numerical setup is assessed. In order to analyse potential discrepancies between the simulated results and experimental measurements, a global energy balance of the simulated results is performed over the entire compartment, which is then contrasted to an experimental balance performed on test I [24].

## 2. Materials and Methods

Details of the numerical solver, of the numerical setup, experimental data, and data comparison are given in this section.

### 2.1. Numerical Solver

FireFOAM is a turbulent diffusion flame solver based on OpenFOAM, an open-source general-purpose CFD software package [22]. FireFOAM is an object-oriented, C++ based, second-order accurate, finite volume method (FVM) solver with implicit time integration [22]. The PIMPLE method, a combination of the semi-implicit method for pressure linked equation (SIMPLE) and pressure-implicit with splitting of operators (PISO) algorithms for pressure–velocity coupling, is used to iteratively solve the governing equations. As stated in Section 1, the solver counts with advanced mesh treating—including both regular and irregular polyhedral structures—and parallel processing capabilities by using message passing interface (MPI) protocols.

The continuum solving is based on the Favre-filtered, LES-based, low Mach number compressible Navier–Stokes equations [17]. Subgrid-scale turbulence is modelled using a single-equation eddy viscosity model, based on formulating the conservation of turbulent kinetic energy (TKE) [17]. Numerical solvers for slow, reactive flows, such as those used for fire scenarios, face the problem that their main driving force results from the buoyancy induced by the temperature differences produced by the exothermic reaction. The constant density assumption, usually invoked for low Mach number flows and which results in incompressible flow (divergence free flow), is thus, not adequate. When only small temperature differences are expected (e.g., in cases of natural convection), the Boussinesq approximation can be used, where a constant density is assumed in the inertial term (and thus the divergence free flow is maintained), but a buoyancy force resulting from temperature differences is introduced [25]. Temperature differences, and the resulting density variations, resulting from combustion, however, are much larger, and the underlying assumptions of the Boussinesq approximation are no longer valid. Rehm and Baum proposed a novel approach for such cases, considering a spatially uniform mean pressure appearing in both the energy equation and the equation of state, and a spatially non-uniform portion of the pressure appearing in the momentum equation, thus allowing for significant density and temperature variations as the pressure remains almost constant in space [26]. The low Mach number solver of FireFOAM uses a similar approach.

Complete combustion of propane is assumed (although the fuel can be, in principle, changed), and is modelled as an infinitely fast chemical reaction, with a fuel reaction rate $\overline{\omega}_{C_3H_8}$ given by the eddy dissipation concept [27],

$$\overline{\omega}_{C_3H_8} = -C_r\overline{\rho}\frac{\varepsilon}{k}\min\left[\widetilde{Y}_{C_3H_8}, \frac{\widetilde{Y_{O_2}}}{s_t}\right] \tag{1}$$

where $C_r$ is a model constant equal to 4 [27]. $\overline{\rho}$, $\varepsilon$, and $k$ are the LES-filtered density, turbulent kinetic energy dissipation rate, and turbulent kinetic energy, respectively. $\widetilde{Y}_i$ represents the Favre-filtered mass fraction of species $i$. Thermal radiation is modelled through the radiative transfer equation (RTE), where a grey medium is assumed, i.e., physical quantities are treated as averaged in the electromagnetic spectrum [17]. The RTE is numerically

solved by implementing the finite volume discrete ordinates method (FVDOM) [28]. For the purposes of this investigation, FireFOAM 4.1 was used to run the simulations.

### 2.2. Experimental Data

Figure 1 shows the ETFT experimental compartment. Only relevant instrumentation will be described herein, and the reader is directed to reference [23] for detailed information about the thermal properties and instrumentation of the compartment.

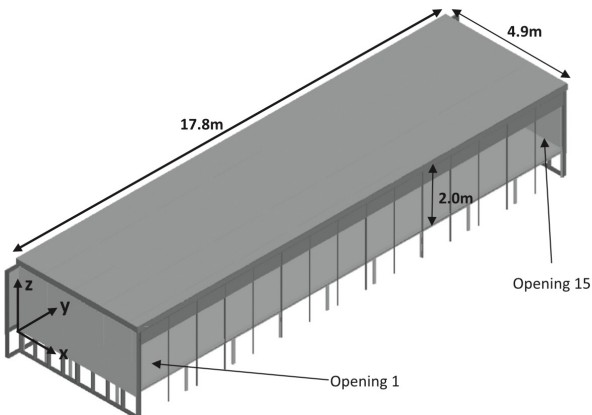

**Figure 1.** Experimental compartment with characteristic internal dimensions and global origin. Source: [23].

Inconel sheathed type K thermocouples with a 1.5 mm bead were used to create temperature maps in the compartment interior and at openings. Internal thermocouples were laid out in a configuration of seven arrays along the *x*-direction (refer to coordinate system in Figure 1), with each array containing 29 thermocouple trees and each tree containing 8 beads. The arrays were labelled A–G along the *x*-direction and 1–29 along the *y*-direction.

Figure 2 shows a plan view of the compartment, with relevant instrumentation represented as symbols. Apart from the internal trees, there was a tree at the centre-line of each of the 15 openings, with five thermocouples per tree. Figure 3 shows a photograph of the 15 openings in the front of the experimental compartment during test I.

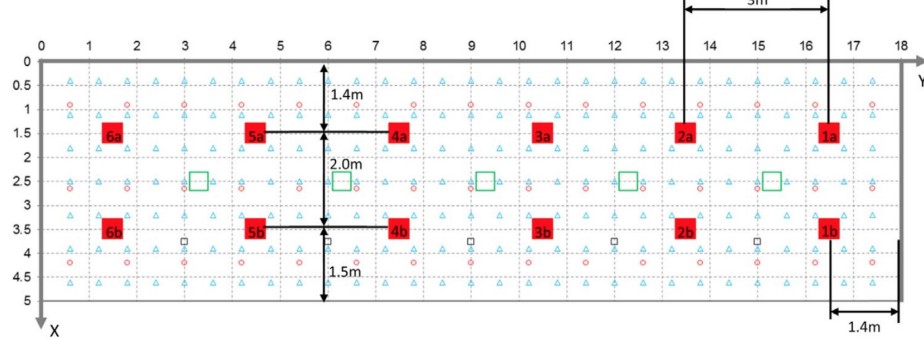

**Figure 2.** Plan view layout of the compartment, showing propane burners (large red squares, located at floor level with pairs labelled 6–1 from **left** to **right**), internal thermocouple trees (blue triangles), floor- and ceiling-level thin skin calorimeters (red circles), gas species sampling points just below ceiling level (small black squares), and obscuration gauges (large green squares, also close to ceiling level). All values are in m and referenced to the global coordinate system. Source: [23].

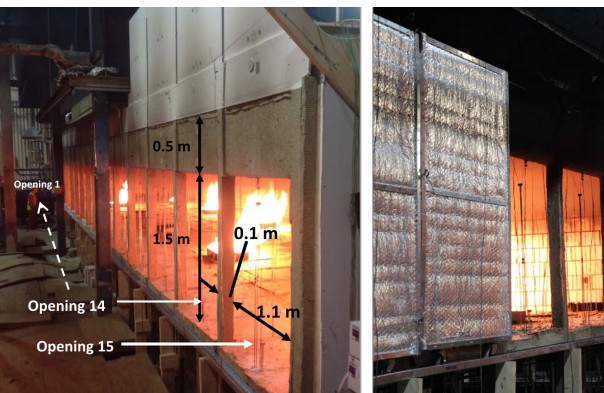

**Figure 3.** Layout and dimensions of the 15 openings along the front (open). Source: [23].

Thin skin calorimeters (TSCs) were used for estimating the incident radiant heat flux. The calorimeters were placed on all five internal surfaces of the compartment. Forty-five TSCs were located on the ceiling of the compartment, arranged in 3 rows of 15 gauges in a mirror-like configuration Figure 2. Forty-five additional gauges were located on the back wall, in a grid of 3 rows of gauges in height, with 15 along each row along the length of the compartment. The right and left walls, corresponding to planes $y = 17.8$ m and $y = 0.1$ m respectively, had 15 gauges each, (3 rows of 5 gauges).

Each of the 15 openings contained two bi-directional velocity probes, one 0.22 m from the base of the opening to capture fresh air inflow velocity, and another one at 1.23 m above the opening base to capture the outflow velocity of hot gases.

### 2.3. Numerical Setup

Both geometry and mesh were built in the open-source software Gmsh [29]. Two different mesh types were considered:

- Structured mesh with rectangular cells.
- Unstructured mesh with tetrahedron cells.

Figure 4 shows the gas-phase domain constructed in Gmsh for both the regular and irregular meshes. Figure 5 shows the composition of the meshes used, from the coarsest to the finest.

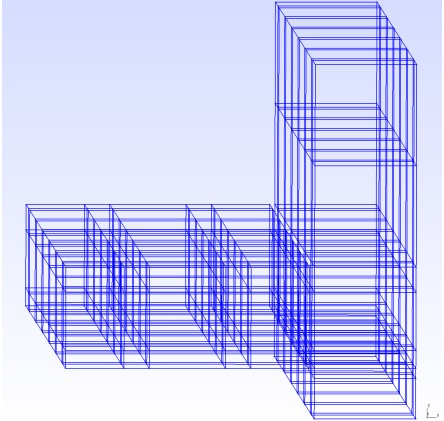

**Figure 4.** Gas-phase domain built for cubic meshing. The left structure corresponds to the compartment interior, and the right structure to the secondary entrainment/escape zone.

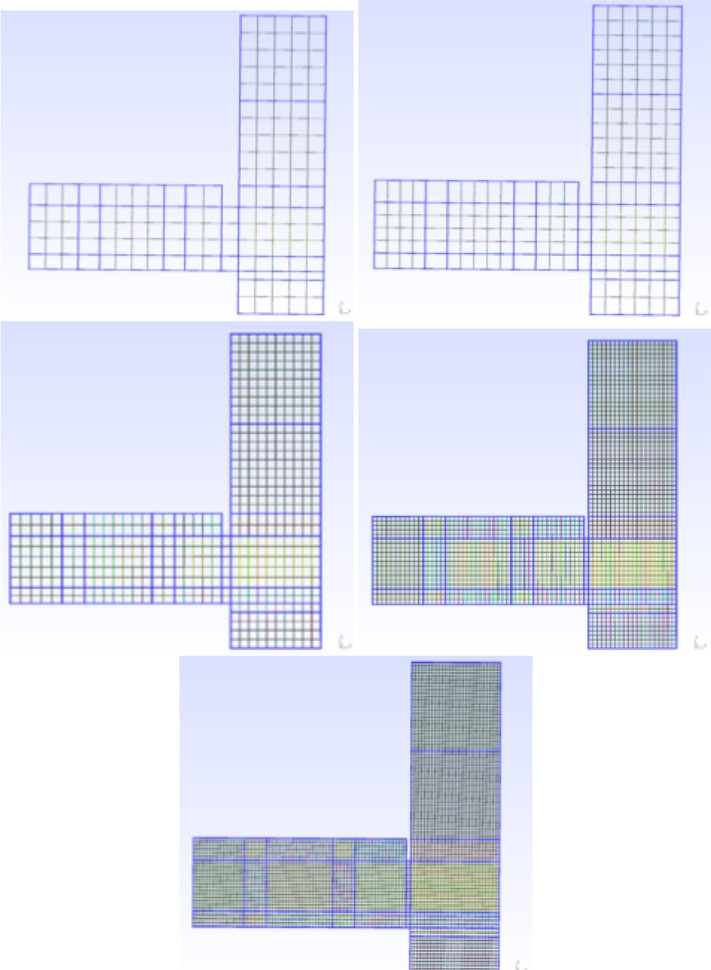

**Figure 5.** Front view of the different cubic meshes. From coarsest to finest: M40, M30, M20, M10, and M7.5.

Spacings between openings—1.5 m high, 10 cm square section profiles, shown in Figure 3—were not included as they are subgrid structures for the 20, 30, and 40 cm meshes and are not expected to cause major disturbances on the velocity fields near the opening centre lines—where probe measurements are made. The hanger, shown in Figure 3 as the hanging 0.5 m structure, was considered to be 10 cm thick ($x$-wise, entering the compartment). This hanger controls smoke accumulation and consequently the smoke-layer height. Finally, a secondary zone was added to the computational domain, adjacent to the compartment openings, in order to capture the air entrainment/smoke escape from/to the rest of the room. The dimensions of this secondary zone, selected as to prevent turbulent structures at boundaries, were 2 m in $x$-span, 17.8 m in $y$-span, and 4 m in $z$-span. Table 1 summarises the fundamental mesh information for each tested case.

The simulations begin when the propane supply to the burners is opened—i.e., when the HRR begins rising, and ends after the ending of 99% of the $HRR_{max}$ steady-state period of almost 2.5 MW. These key events are graphically shown in Figure 6. Figure 6 also shows that the simulation captures three steady-state periods: 50% $HRR_{max}$, 74% $HRR_{max}$, and 99% $HRR_{max}$. The timestep is adjusted by the solver in each step as to maintain the Courant number under unity in the entire domain, which assures numerical stability.

**Table 1.** Elements and vertices reported post-meshing by Gmsh. All numbers approximated to the third significant figure. 'M' represents regular meshing, and 'T' irregular (or tetragonal). The number after the letter indicates the average cell size in cm.

| Case | Number of Vertices | Number of Elements |
|------|--------------------|--------------------|
| M40 | 7.38 k | 18.1 k |
| M30 | 13.7 k | 29.0 k |
| M20 (B) | 50.2 k | 82.9 k |
| M10 | 432 k | 558 k |
| M7.5 | 943 k | 1152 k |
| T20 | 85.7 k | 501 k |

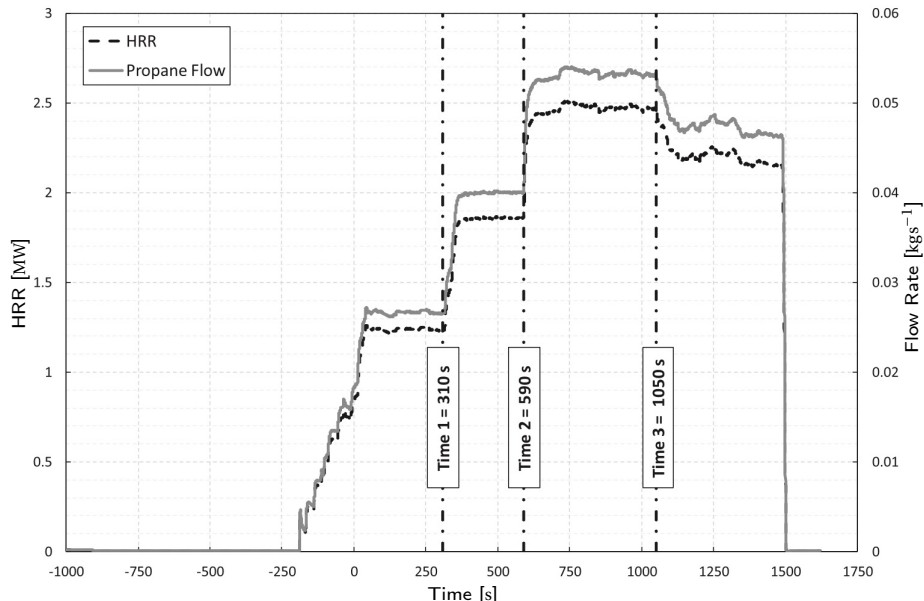

**Figure 6.** Propane flow rate from the large mass flow controller (solid line) and corresponding HRR assuming 100% combustion efficiency within the compartment (horizontally dashed line). The vertical dashed line marks propane mass flow changes. Source: [23].

The main boundary conditions are summarised in Table 2. Air entrainment through the outlet and side entrainment surfaces can be modelled through the combination of the boundary conditions *pressureInletOutletVelocity* and *totalPressure* for velocity ($\widetilde{\mathbf{u}}$) and modified pressure ($\overline{p_m}$), respectively [17]. The modified pressure ($\overline{p_m}$) is defined in OpenFOAM as:

$$\overline{p_m} = \overline{p} - p_{ref} - \overline{\rho}(\mathbf{g} \cdot \mathbf{x}), \tag{2}$$

where $\overline{p}$, $p_{ref}$, $\mathbf{g}$, and $\mathbf{x}$ are the Favre-averaged thermodynamic pressure, reference pressure (101.325 kPa), gravity acceleration vector, and cell location vector, respectively. Using this modified pressure has been proven to improve the numerical solution [17]. OpenFOAM-4.1 has implemented *prghTotalHydrostaticPressure* as a substitute to the *totalPressure* boundary condition for the modified pressure ($\overline{p_m}$). This $\overline{p_m}$ boundary condition uses the hydrostatic pressure field as the reference state for the far field, which provides more accurate entrainment in large open domains, typical of many fire scenarios. Regarding inlet and wall patches, the modified pressure was established through the *fixedFluxPressure* boundary condition, adjusting its gradient so that it yields the flux given by the velocity boundary condition.

**Table 2.** Main OpenFOAM boundary conditions. Propane inflow is set according to a linear piece-wise function adjusted to the experimental measurement of the large mass flow controller. Atmospheric conditions assumed: $p^\infty = 101.3$ kPa, $T^\infty = 283.65$ K, $k^\infty = 10^{-4}$ m$^2$ s$^{-2}$, $Y_{N_2}^\infty = 0.767$, $Y_{O_2}^\infty = 0.233$.

| | $\widetilde{\mathbf{u}}$ | $\overline{p_m}$ |
|---|---|---|
| Outlet | pressureInlet OutletVelocity | prghTotal HydroPress |
| Entrainment | '...' | '...' |
| Burners | flowRate InletVelocity | fixed FluxPress |
| Burner walls | noSlip | '...' |
| Hanger | '...' | '...' |
| Floor/roof/walls | '...' | '...' |
| Room walls | '...' | '...' |
| | $\widetilde{Y_i}$ | $k$ |
| Outlet | inletOutlet: $Y_i^\infty$ | intetOutlet: $k^\infty$ |
| Entrainment | '...' | '...' |
| Burner | totalFlowRate AdvDiff | fixedValue: $k^\infty$ |
| Burner walls | zeroGradient | '...' |
| Hanger | '...' | '...' |
| Floor/roof/walls | '...' | '...' |
| Room walls | '...' | '...' |
| | $\widetilde{T}$ | $I$ |
| Outlet | inletOutlet: $T^\infty$ | greyDiff Rad: 1 |
| Entrainment | '...' | '...' |
| Burner | fixedValue: $T^\infty$ | greyDiff Rad: 0.9 |
| Burner walls | zeroGradient | '...' |
| Hanger | '...' | '...' |
| Floor/roof/walls | '...' | '...' |
| Room walls | '...' | '...' |

For species inlet at the burner, the *totalFlowRateAdvectiveDiffusive* boundary condition was used with a mass fraction equal to one for propane and zero for the rest. Combined with the velocity boundary condition *flowRateInletVelocity*, which defines mass flow rate as a function of time, it is assured that the prescribed mass flow rate accounts for both advection and diffusion fluxes. Prescribing a fuel mass flow rate which does not account for diffusion will cause an over-estimation of the heat release rate, which is particularly important in turbulent buoyant diffusion flames where fuel concentration gradients near the inlet are steep, inducing an additional flow [22]. This methodology is used in conventional diffuser burner [17] and porous burner [19] configurations, the latter corresponding to burners used in ETFT test I. As the propane flow rate was directly measured in test I, the replication of this boundary condition—and hence the total HRR—is expected to be accurate. Figure 7 shows both experimental and boundary-condition-prescribed mass flow rates. For all species, an *inletOutlet* boundary condition was established at the *outlet* and *entrainment* patches, which imposes *zeroGradient* for outflow and *inletValue* for inflow. As seen in Table 2, in the case of inflow through the *outlet* or *entrainment* surfaces, volume fractions of 0.767 for N$_2$ and 0.233 for O$_2$ were prescribed for air when entrained into the domain.

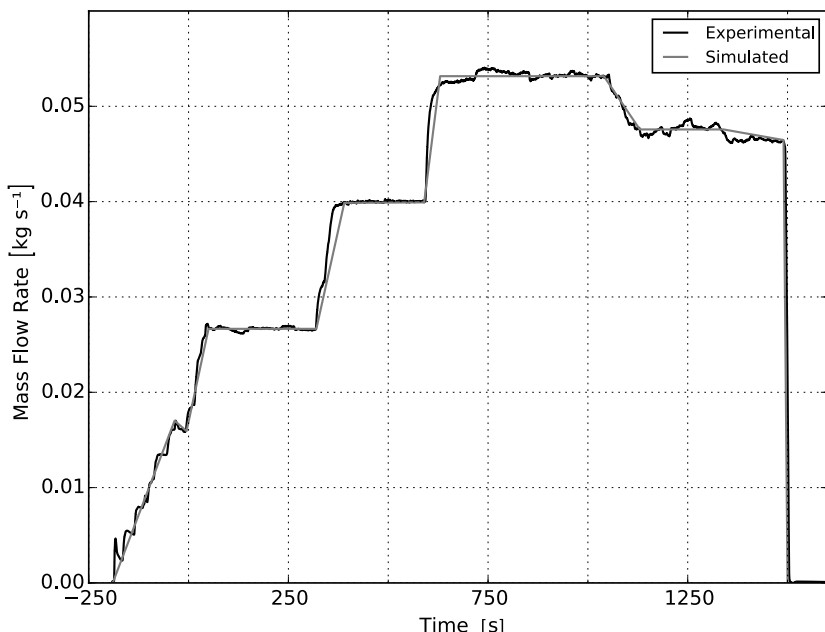

**Figure 7.** Total prescribed mass flow rate (advective plus diffusive) at burner inlet patches, built as a piece-wise function of linear functions.

Considering the implementation of the radiation FVDOM model, the corresponding *MarshakRadiation* and *greyDiffusiveRadiation* boundary conditions were established for incident radiation ($G$) and radiation ray intensity ($I$), respectively. The outlet and side entrainment zones were considered as black bodies (i.e., $\varepsilon = 1$), a typical assumption for construction materials. All the constructive materials at the compartment interior were considered to have an emissivity equal to 0.9, following the recommendation proposed by Maluk et al. for the ETFT series [24].

Soot formation/oxidation is neglected, as complete soot formation/oxidation modelling is only justifiable in scenarios with highly sooting fuels, or with large pool fire diameters with crosswinds (e.g., large-scale LNG sea pool fires) where limited air entrainment at the centre of the pool and wind distributions promote uneven soot distribution [30]. Soot influence was exclusively considered in the radiation model, considering a simple *mixtureFractionSoot* already implemented in FireFOAM. Only $C_3H_8$, $H_2O$, and $CO_2$ participate in radiation, as $O_2$ and $N_2$ are diatomic molecules and hence do not present strong emission bands in the infrared region of the electromagnetic spectrum [17]. Soot is only considered to participate in radiation scattering as a constant soot fraction, according to the chemical reaction of combustion.

As no reference values were available for propane diffusion flames, the 0.055 constant was taken from the methane-fuelled *smallPoolFire2D* OpenFOAM tutorial case. As propane is a medium-sooting fuel, it was deemed preferable to include an under-estimated soot fraction—corresponding to the value for methane—rather than no soot fraction.

Finally, for temperature boundary conditions, a prescribed inlet temperature equal to ambient conditions—283.65 K (10.5 °C)—, was considered for the fuel, neglecting inlet fuel heating due to flame radiation. Heat dissipation through walls was modelled through the *externalWallHeatFluxTemperature* boundary condition, which uses a steady-state 1D heat transfer model that considers internal convection, internal wall conduction, and external convection with ambient air at 283.65 K (10.5 °C). The radiation field ($Q_r$) was not considered in the energy balance as it caused numerical instability dependent on the solver's relaxation factor. The former was solved considering an equivalent internal convection coefficient $h'_w$ that includes radiation, which was calculated using the original average convection coefficient from the compartment interior walls, $h_w = 9.5 \ \mathrm{W \, m^{-2} \, K^{-1}}$ [24], and experimental measurements of temperature and incident radiation [23]. A value of

$h'_w = 21 \, \text{W} \, \text{m}^{-2} \, \text{K}^{-1}$ was obtained. This value is approximately 2.2 times higher than the original convection coefficient, indicating that radiation accounts for 55% of the total wall heat loss (with convection accounting for the other 45%). These numbers are in good agreement with vertical wall fire heat distribution studies [19], where it was found and validated with the literature and correlations that in the inert wall region convection and radiation accounted for approximately 40% and 60%, respectively. The slightly higher radiation proportion can be justified by the fact that in vertical wall fires, the flame is attached to the wall and, therefore, the entire wall surface faces the flame. Thermal conduction coefficients and material composition for the floor, internal walls, and roof were obtained from [23]. The external convective coefficient ($h_{ext} = 20 \, \text{W} \, \text{m}^{-2} \, \text{K}^{-1}$) was taken from the Chilean HVAC Standard NCh 853/2007 [31].

*2.4. Data Comparison*

The simulated temperatures were compared to the thermocouple measurements. In turbulent diffusion flames, where thermal radiation is important, thermocouple temperatures (raw measurements) will present significant discrepancies with respect to the gas-phase temperatures. It is expected that close to the ceiling, where the compartment accumulates hot gases and receives considerable amounts of flame radiation, the radiations levels will be high and, therefore, the thermocouples will read higher temperatures than the adjacent gas phase. This is expected to produce differences of 10–100 K [23]. Thus, simulated gas-phase temperatures ($T_g$) will be used in order to calculate thermocouple temperatures ($T_{tc}$), which will be more suitable for comparison with raw thermocouple data. Performing an energy balance in the thermocouple bead, the thermocouple model results in [19]:

$$\rho_{tc} c_{tc} \left( \frac{V_{tc}}{A_{tc}} \right) \frac{dT_{tc}}{dt} = \varepsilon_{tc}(G - \sigma T_{tc}^4) + h_{tc}(T_g - T_{tc}), \tag{3}$$

where $\rho_{tc}$, $c_{tc}$, $V_{tc}$, $A_{tc}$, and $\varepsilon_{tc}$ are the thermocouple properties: mass density, specific heat capacity, volume, surface area, and emissivity, respectively. Thermocouples were modelled as 1.5 mm nickel spheres [23], with density and specific heat capacity of 9000 kg m$^{-3}$ and 440 J kg$^{-1}$ K$^{-1}$ [32], respectively. The thermocouple surface emissivity was considered to be 0.9 [32]. Equation (3) was iteratively solved, as the external convective coefficient ($h_{tc}$) depends on the simulated thermocouple temperature ($T_{tc}$). This coefficient was calculated considering both forced and natural external convection correlations [33], depending on the simulated Reynolds number.

Raw data from thin skin calorimeters (TSCs) is reported as the disc temperature, and data processing as indicated by Hidalgo et al. was performed [34]. The procedure is based on an energy balance performed in the TSC disc.

Additionally to point comparison of temperatures and heat fluxes, an energy balance based on simulated data is reported in this study, with the purpose of verifying that FireFOAM is both conservative and capable of reproducing the experimental energy flows for test I reported by Maluk et al. [24]. The heat release rate ($\dot{Q}_{\text{fire}}$) is expected to be equal to the sum of the heat rate stored by the gas-phase ($\dot{Q}_{\text{gas}}$) net enthalpy flux leaving the opening plane ($\dot{H}_{\text{out,op}} - \dot{H}_{\text{in,op}}$), heat dissipation through walls ($\dot{Q}_{\text{walls}}$), and the radiative flux leaving through the compartment opening plane ($\dot{Q}_{\text{rad,op}}$):

$$\dot{Q}_{\text{fire}} = \dot{Q}_{\text{gas}} + (\dot{H}_{\text{out,op}} - \dot{H}_{\text{in,op}}) + \dot{Q}_{\text{walls}} + \dot{Q}_{\text{rad,op}}, \tag{4}$$

where each term is calculated discretely considering simulated data and numerical mesh dimensions:

$$\dot{Q}_{\text{gas}} = \sum_{\text{cells}} V_i \cdot \rho(T_i) \cdot c_p(T_i) \cdot \frac{\Delta T_{g,i}}{\Delta t}$$

$$\dot{H}_{\text{out,op}} - \dot{H}_{\text{in,op}} = \sum_{\text{op. patches}} \rho(T_{g,i}) A_i u_x \cdot c_p(T_{g,i}) \cdot T_{g,i}$$

$$\dot{Q}_{\text{walls}} = \sum_{\text{wall patches}} U_{\text{walls}}(T_{g,i} - T_{\text{amb}}) +$$

$$\sum_{\text{roof patches}} U_{\text{roof}}(T_{g,i} - T_{\text{amb}}) + \sum_{\text{floor patches}} U_{\text{floor}}(T_{g,i} - T_{\text{amb}})$$

$$\dot{Q}_{\text{rad,op}} = \sum_{\text{op patches}} G_i \cdot A_i$$

Global heat transfer coefficients ($U_i$) were used, which is consistent with the steady-state 1D wall boundary condition. The coefficients were calculated considering the equivalent internal convective heat transfer coefficient—which considers radiation in the balance, external convective coefficient, and solid thermal layer lengths ($L_j$) and conductivities ($\gamma_j$):

$$U_i = \left[ \frac{1}{h'_w \cdot A_p} + \sum_j \frac{L_j}{\gamma_j \cdot A_p} + \frac{1}{h_{ext} \cdot A_p} \right]^{-1} \tag{5}$$

Regarding the mass balance, discrete mass conservation is applied at the boundaries of the entrainment/escape zone of the numerical domain. The following expression is solved:

$$\frac{\partial}{\partial t}(\bar{\rho}_a \cdot \Delta V) + \sum_{f=1}^{6} \bar{\rho}_f (\widetilde{u}_f \cdot \hat{n}_f) S_f = 0, \tag{6}$$

where $S_f$ and $\hat{n}_f$ correspond to the surface area of face $f$ and its normal vector. $\bar{\rho}_f$ and $\bar{u}_f$ are the face-average density and velocity vector, and $\bar{\rho}_a$ is the cell-average density.

## 3. Results

Before comparing physical quantities, the performance of FireFOAM in terms of run-time vs. mesh size is assessed, and the results are reported. Then, the overall mass and energy balance is presented. Finally, temperatures and heat fluxes are compared. Particularly, the validation process presented in this study focuses on comparing time-averaged thermocouple temperatures, time-averaged back wall incident radiation, and time-averaged inflow and outflow. Additionally, time-averaged streamlines are plotted to illustrate and assess the internal flow field.

### 3.1. Run-Time Sensitivity

A host computer with eight Intel(R) Core(TM) i7-4820k CPU @3.70 GHz processors, and 64 GB RAM memory was used for computing the results.

The performance study for different mesh sizes (Figures 8 and 9) is presented with a logarithmic scale in order to capture all orders of magnitude. As expected, the run-time is significantly sensitive to the mesh size. As the mesh size decreases, not only has the solver to compute a larger amount of cells, but it has to do this more often, as smaller timesteps result from the fact that the Courant number has to be kept lower than unity. Considering this, only cases M30 and M40 run faster than real time, where M30 has a small advantage of a few seconds (approximately 70 s). Note that the T20 case (irregular tetrahedron mesh) showed numerical divergence and hence was not included in the comparison of Section 3.

It is important to note that, due to the mesh construction algorithm which keeps the original compartment dimensions intact, the mesh size only represents an approximation of the average cell dimension. As the cell dimension approximates the structural characteristic dimensions, e.g., 50 cm for the burner, coarser meshes imply a small change in the number of elements. For example, refinement from M40 (18 K elements) to M30 (29 K elements) only increases the number of elements by less than two times, while from M20 (83 K elements) to

M10 (560 K elements) the number of elements increases almost seven times. This explains the big increase in run-time when switching from M20 to M10.

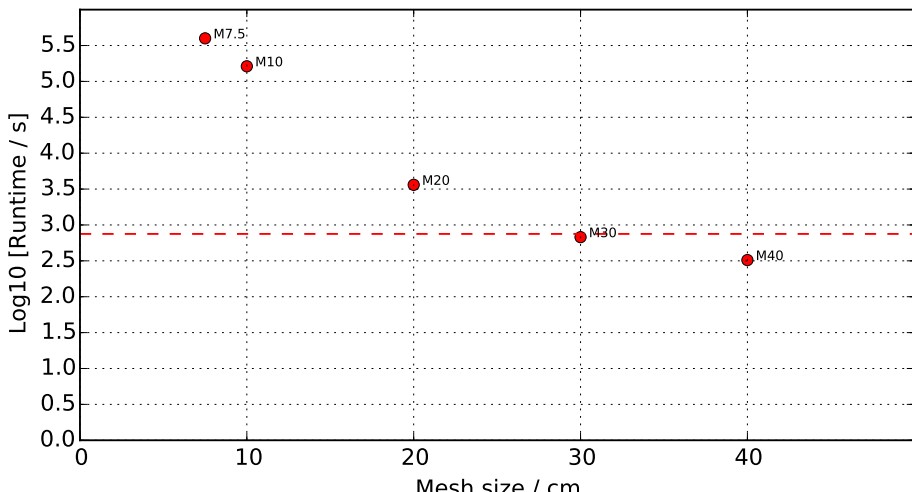

**Figure 8.** Run-time for different meshes. Simulations are run starting from the propane supply to the burners, and ending at the 74% HRR steady-state period. A log10 scale is applied for run-times to capture all magnitude orders. The red dashed line marks the real-time limit, i.e., log10(750 s) = 2.88.

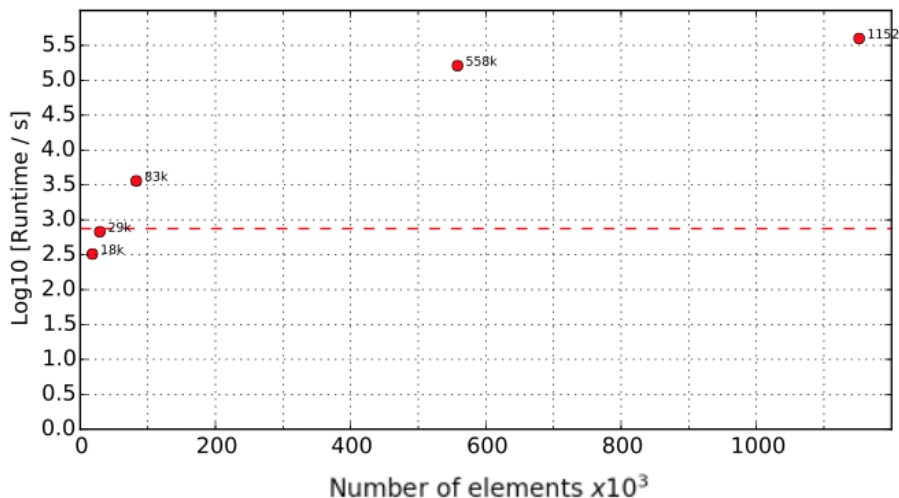

**Figure 9.** Run-time for cases M40, M30, M20, M10, and M7.5 represented as number of elements rather than average cell size. The red dashed line marks the real-time limit, i.e., log10(750 s) = 2.88.

The effect of the element number on the run-time is shown in Figure 9. Further refining high-resolution meshes, e.g., from M10 to M7.5, involves a large increase in the number of elements, as these are much smaller than compartment structures. This is a characteristic of the designed meshing algorithm which has to be considered when interpreting the results.

Significant speed-up due to parallel processing is shown in Figure 10, particularly for the first parallel processors. When increasing from 1P to 2P, the run-time decreases by 4000 s, while form 2P to 4P it decreases only by 2400 s. Further addition of parallel processors is not beneficial past the fourth processor. This is deemed to be due to excessive message passing between processors. The parallel efficiency parameter quantifies how message passing becomes highly interrupted past this point: with six processors, less than half of the total run-time (41%) is used for computing results, while the rest (59%) is used in message passing.

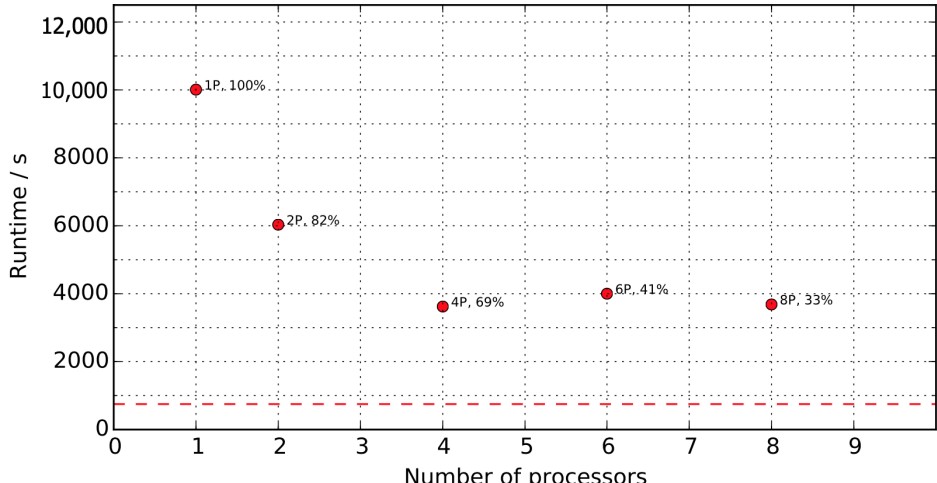

**Figure 10.** Run-time for different numbers of parallel processors. Parallel efficiency, expressed as a percentage, is calculated as $\epsilon = \frac{T_1}{p \cdot T_p}$ (where $T_p$ is the computing time for $p$ processors) and placed adjacent to each case. The red dashed line marks the real-time limit, i.e., 750 s. M20 case.

Figure 11 shows the run-time as a function of the number of solid angles. In the range of 4 to 144 solid angles, the increase in run-time is relatively low—only approximately 550 s of real time—compared to the increase in the range of 144 to 400 solid angles. From 144 to 256 solid angles a 3400 s increase in run-time is observed, while from 256 to 400 solid angles the run-time increases by 2000 s. As the later results indicate, incident radiation convergence with solid angles is achieved at approximately 16 solid angles for the M20 case back wall, and using from 16 to 144 solid angles is, thus, reasonable for the baseline. This number is expected to be mesh-dependent, as will be discussed later on.

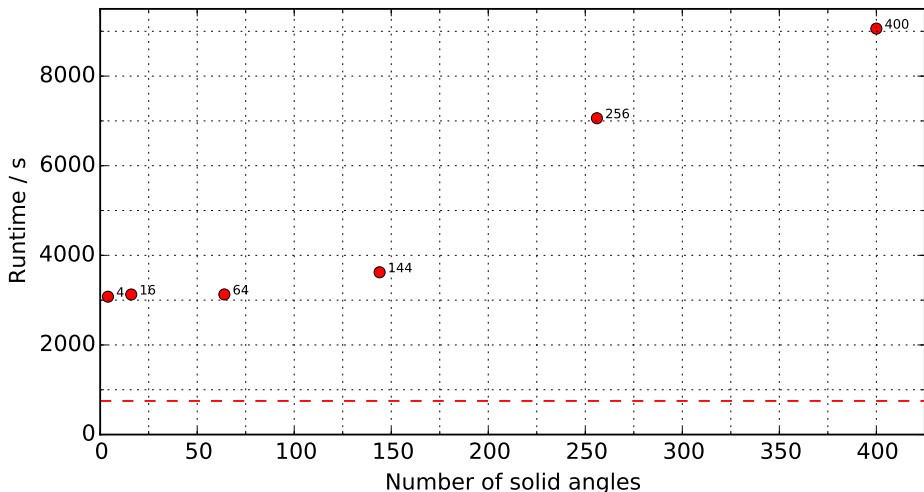

**Figure 11.** Run-time for different number of solid angles. A 20 cm cubic mesh is used in all cases. The red dashed line marks the real time limit, i.e., 750 s.

### 3.2. Mass Balance

An overall mass balance is shown in Figure 12 for the M20 and M10 cases. It is expected that the net mass outflow is zero during all steady-state periods, considering that during these periods the compartment does not lose/gain gas-phase mass and the fuel flow rate is negligible compared to the air entrainment. This was not accomplished for either the M20 or M10 cases, as a net outflow of $0.5 \text{ kg s}^{-1}$ is observed in both cases. This discrepancy represents approximately 7% of the maximum outflow of $7.5 \text{ kg s}^{-1}$, which is above the expected calculation error (calculation error is understood as deviations from expected

values due to the summed effect of interpolation, differentiation, and integration errors (numerical error)) of approximately $\pm 5\%$. As the $0.5\,\mathrm{kg\,s^{-1}}$ discrepancy is equal for both the M10 and M20 cases, it is attributed to systematic errors resulting from imposing models on governing equations (e.g., subgrid-scale modelling). The introduction of subgrid-scale modelling promotes deviations from the conservative form of the Navier–Stokes system, as shown in [17]. Inflow and outflow were calculated at the compartment opening plane, as all other internal patches were either walls or burners.

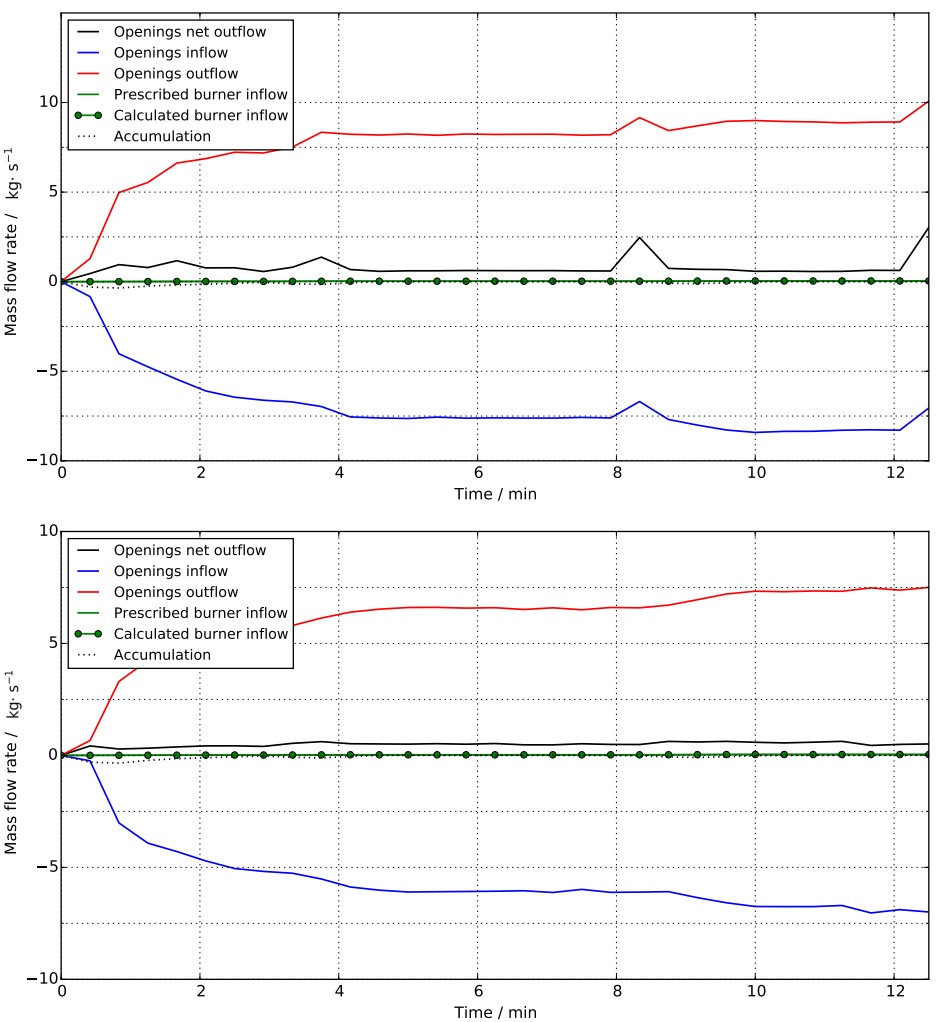

**Figure 12.** Mass fluxes considering the compartment interior as a control volume. (**Up**): M20. (**Down**): M10.

Finally, both cases show consistency regarding mass loss inside the compartment during transient periods. Figure 12 captures this phenomenon as negative accumulation (black dotted curves) that can be seen when the heat release rate increases. This is physically consistent: as the heat release rate increases, internal temperatures also increase and, therefore, the compartment volume holds less mass (due to a mean density decrease). Negative mass accumulation periods account for this extra mass leaving the compartment.

*3.3. Energy Balance*

Figure 13 shows the normalised energy balance for the M10 case. The baseline M20 case was not included as discontinuities were identified in the energy balance, particularly in the calculated HRR and net enthalpy outflow, and further discussions regarding the energy balance will, thus, refer to M10 only.

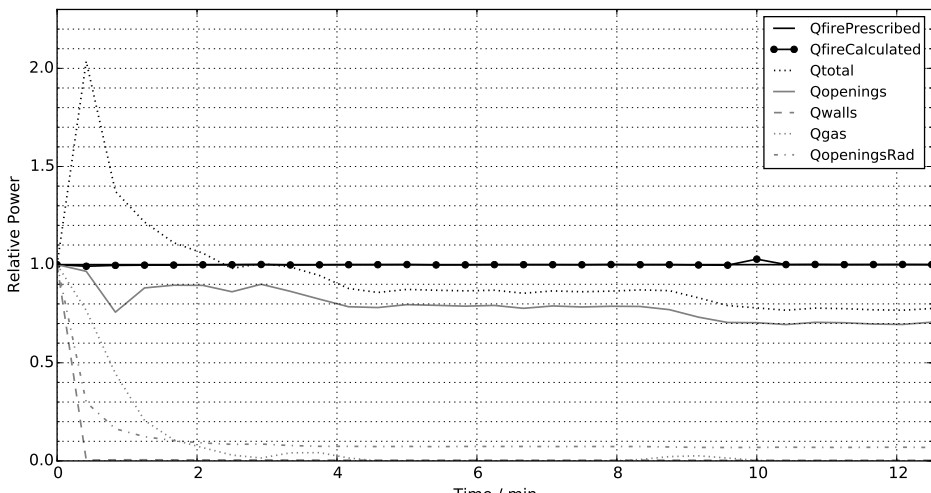

**Figure 13.** Normalised energy fluxes with respect to the prescribed heat release rate, noted in the legend as *QfirePrescribed*.

The prescribed heat release rate was calculated as the prescribed propane mass flow rate multiplied by its specific heat of combustion (46.45 MJ kg$^{-1}$), while the calculated heat release rate was calculated as the sum of the heat released in each compartment cell as calculated by the EDC reaction model. A consistent result is seen in the M10 case, as prescribed and calculated values are equal for all timesteps. The M20 case showed considerable discontinuities between these two values in 4 out of 31 timesteps, with overestimations of at least 10% in the calculated HRR.

The curve denominated *Qtotal* in Figure 13, which represents the sum of energy dissipation through the walls, openings (due to both net enthalpy outflow and radiation), and accumulation as internal energy, should be equal to the heat release rate at all timesteps, considering an approximate ±5% of calculation error. Regarding the first 2 min of the M10 case, consisting of a linear fuel mass flow rate ramp-up, *Qtotal* almost doubled the total heat release rate, mainly because of high values for both *Qgas* and *QopeningsRad*. From minutes 2 to 4, consisting of a brief propane flow rate hold period followed by a ramp-up, *Qtotal* becomes similar to *Qfire* considering the ±5% calculation error. From 4 to 8.5 min, corresponding to 50% of the HRR$_{max}$ steady-state period, *Qtotal* is approximately 85–90% of the total heat release rate, decreasing to almost 80% during the steady-state period with 74% of the HRR$_{max}$ period (minutes 9.5 to 12.5). Therefore, as the heat release rate increases, the unaccounted proportion of energy fluxes becomes higher. This could be due to a flux not accounted for (which becomes higher with increased HRR), or due to deviations from the conservative form of Navier–Stokes due to the introduction of sub-models and model assumptions (e.g., subgrid-scale modelling or others).

The characteristic heating times and proportions of each dissipation type are evaluated next. The characteristic heating time can be quantified in the HRR ramping from 50% of HRR$_{max}$ to 74% of HRR$_{max}$. As seen for the M10 case in Figure 13, this characteristic time is equal to 2 min and occurs between minutes 8 and 10, as *Qgas* rises to a peak of approximately 3–4% of *Qfire* and then goes back to zero. The experimental energy balance study performed by Maluk et al. also reports this characteristic time as 2 min [24], with *Qgas* peaking at approximately 5% of *Qfire* for the 50–74% HRR$_{max}$ transition. Therefore, good agreement between simulation and experiment is seen in the average transient modelling.

Table 3 summarises the simulated (M10) and experimental energy distributions for both the 50% and 74% HRR$_{max}$ steady-state periods. According to the experimental analysis performed in [24], an increase in the heat release rate from 50 to 74% of HRR$_{max}$ increases the proportion of energy dissipated due to hot gas enthalpy leaving the compartment (from 0.67 to 0.7), lowering heat dissipation through the walls (from 0.23 to 0.18). The simulations do not capture this experimental observation, because even though the hot

gas enthalpy flux increases, its proportion relative to *Qfire* decreases from 0.80 to 0.70. Heat dissipation through the internal walls is quantified as less than one percent in both simulations, and given that is quantified as nearly 20% in the experimental case, the heat transfer model, which consists of steady-state one-dimensional heat transfer into each boundary patch, is clearly not representative. The calculated *U*-values for the internal walls, roof, and floor were 0.52 W m$^{-2}$ K$^{-1}$, 0.12 W m$^{-2}$ K$^{-1}$, and 0.26 W m$^{-2}$ K$^{-1}$, respectively. The *U*-values reported in [23] for the internal walls, roof, and floor were 0.18 W m$^{-2}$ K$^{-1}$, 0.15 W m$^{-2}$ K$^{-1}$, and 1.02 W m$^{-2}$ K$^{-1}$, respectively. As the *U*-values in the simulations are of similar magnitudes to the *U*-values reported in [23], especially for the roof, where most of the dissipation occurs, the thermal properties of the boundaries were considered to be well reproduced. Maluk et al. [24] could not experimentally estimate the radiation dissipation through openings, therefore, the unaccounted for 10% of the energy flux (refer to Experimental in Table 3) was attributed to this dissipation mechanism. This experimental estimation of *QopeningsRad* is similar to the simulated value of 8%, which indicates adequate radiation modelling.

**Table 3.** Energy distribution analysis for both steady-state periods.

|  | HRR | 'Qtot' | 'Qop' | 'QopRad' | 'Qwalls' |
|---|---|---|---|---|---|
| **Sim** | 50% | 0.88 | 0.80 | 0.08 | <0.01 |
|  | 74% | 0.78 | 0.70 | 0.08 | <0.01 |
| **Exp** | 50% | 0.90 | 0.67 | N/A | 0.23 |
|  | 74% | 0.88 | 0.70 | N/A | 0.18 |

*3.4. Internal Flow Streamlines*

In order to assess FireFOAM's performance in terms of flow pattern convergence with mesh refinement, steady-state streamline maps for the compartment interior were plotted (Figure 14). Due to the lack of flow data from inside the compartment (velocity probes were placed only at openings), it is not possible to compare streamlines quantitatively to the experiments, and only a qualitative assessment is shown. All cases capture the physical phenomena of air entrainment through the lower opening section and hot gas escape through the upper section. Refining the mesh from case M40 to M30 causes the central recirculation seen in M40 to disappear, but it reappears in case M20. A secondary recirculation appears in M10, and a further refinement to M7.5 does not significantly change the flow pattern that now consists of two steady-state vortex structures. This seems to indicate mesh convergence from M10 onward. A similar outflow velocity modulus of approximately 1.5 m s$^{-1}$ is seen in both M10 and M7.5, which also supports the claim that the M10 case is flow-converged.

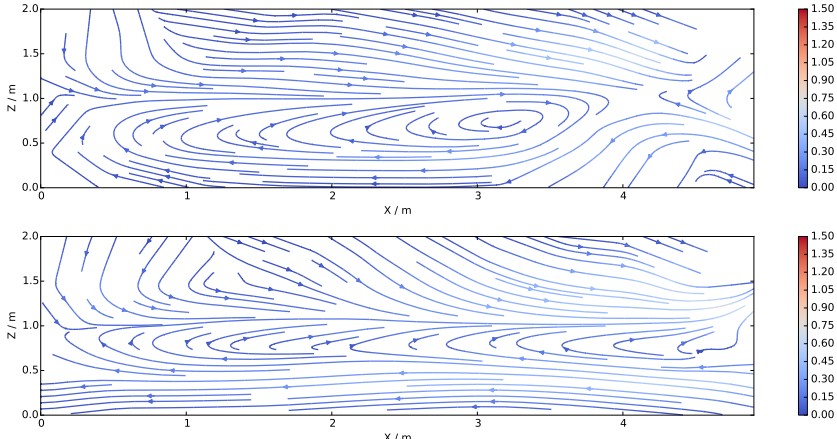

**Figure 14.** *Cont*.

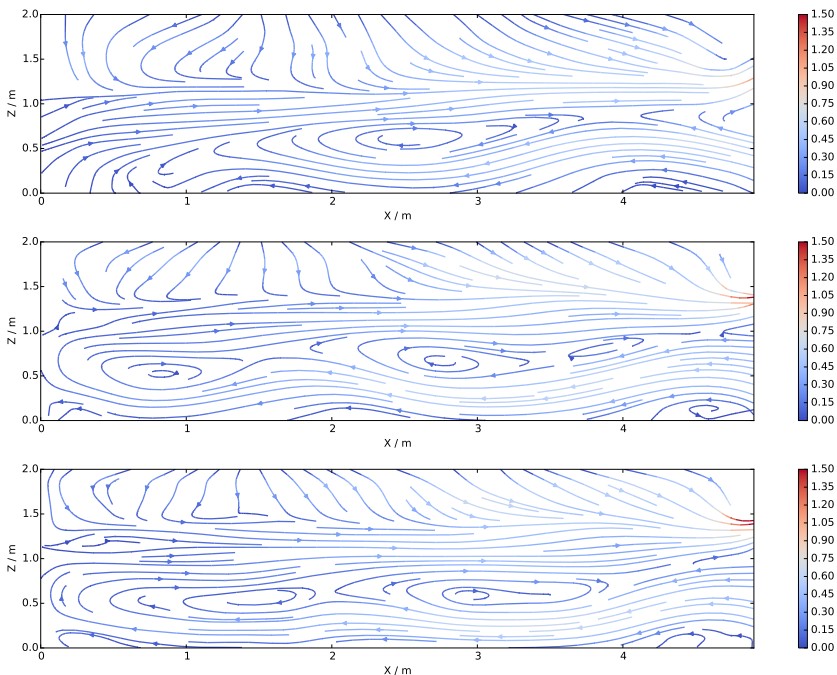

**Figure 14.** Time-averaged streamline maps in section 15 for the steady-state period of 74% HRR. Velocity modulus in m/s. From **top** to **bottom**: M40, M30, M20, M10, and M7.5.

### 3.5. Temperature Contours

Figure 15 shows experimental and simulated temperature contours. The solver captures the experimental neutral plane height of 1–1.2 m [23] for all cases from M20 to M7.5. According to [23], this interface is created by the presence of the hanger. Coarser meshes tend to over-estimate simulated thermocouple temperatures at the ceiling level, mainly due to high estimations of incident radiation (*G*-field).

Uneven distribution of propane among the burners caused elevated temperatures near the far-right pair of burners. Burner 1a exhibited a higher output than the others [23]. This behaviour could not be reproduced in the simulations as the magnitude of the imbalance is unknown. This considered, cases M10 and M7.5 tend to capture the 200 °C hot layer temperature for the 74% HRR$_{max}$ steady-state period better, with over-estimated values of 250–300 °C. Uneven propane distribution can account for some over-estimation error. Considering the section B experimental contour, temperatures in zones aligned with burner pairs 6–2 are approximately 200 °C, while the zone aligned with burner pair 1 exhibits approximately 450 °C. Weight-averaging these temperatures considering each burner pair as a weight gives an estimate of 240 °C in the hot gas layer if even propane distribution had been present in test I. Therefore, the measured smoke-layer temperature from burners 6 to 2 stays approximately 40 °C lower with respect to an even propane distribution scenario.

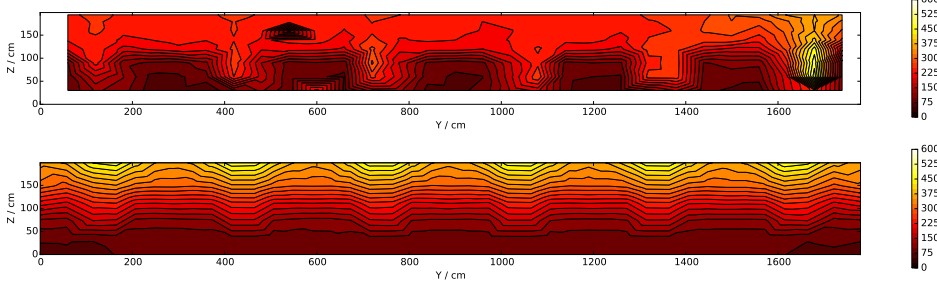

**Figure 15.** *Cont.*

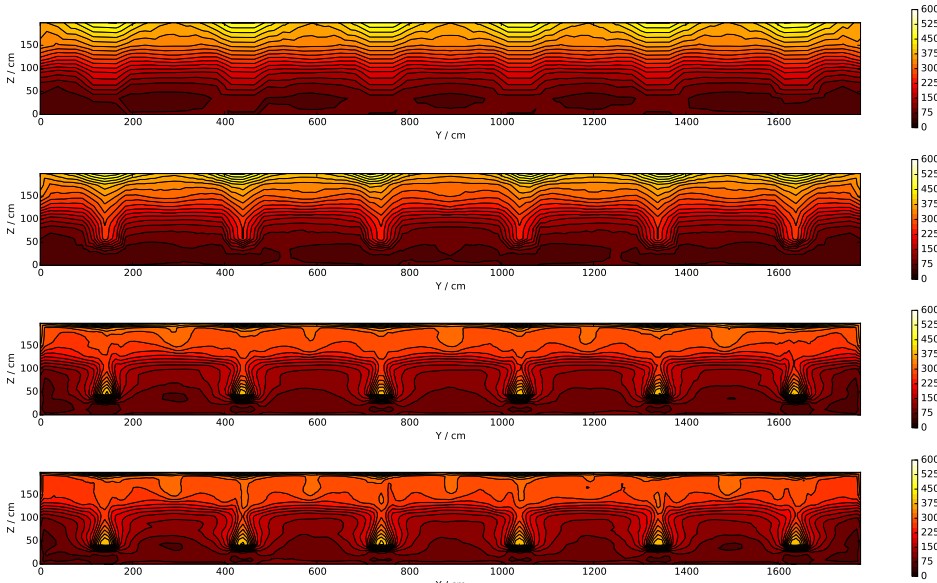

**Figure 15.** Time-averaged thermocouple temperature contours in section B for the steady-state period of 74% HRR. Temperatures in °C. From **top** to **bottom**: Experimental, M40, M30, M20, M10, and M7.5.

## 3.6. Incident Radiation Contours

Figure 16 shows the results for incident radiation (*G*) convergence with increasing number of solid angles in FVDOM, considering the M20 case. Note that the simulated cases are reported with a wider scale compared to the experimental case (0–20 kW m$^{-2}$ in simulations vs. 0–10 kW m$^{-2}$ in the experiments), indicating an important over-estimation of incident radiation in the hot gas layer. Regarding convergence with solid angles, the M20 case is converged with only 16 solid angles. This value is much smaller than the 300-solid-angle convergence reported in [18] for FireFOAM. This discrepancy is attributed to the difference in mesh sizes, as the cited study uses 2–12 mm cubic meshes, compared to the cubic 200 mm mesh used here. The number of solid angles needed for *G* convergence might thus increase with finer meshes. It was found that further improvement in the *G* estimation is achieved through finer meshes rather than more solid angles, as Figure 17 shows.

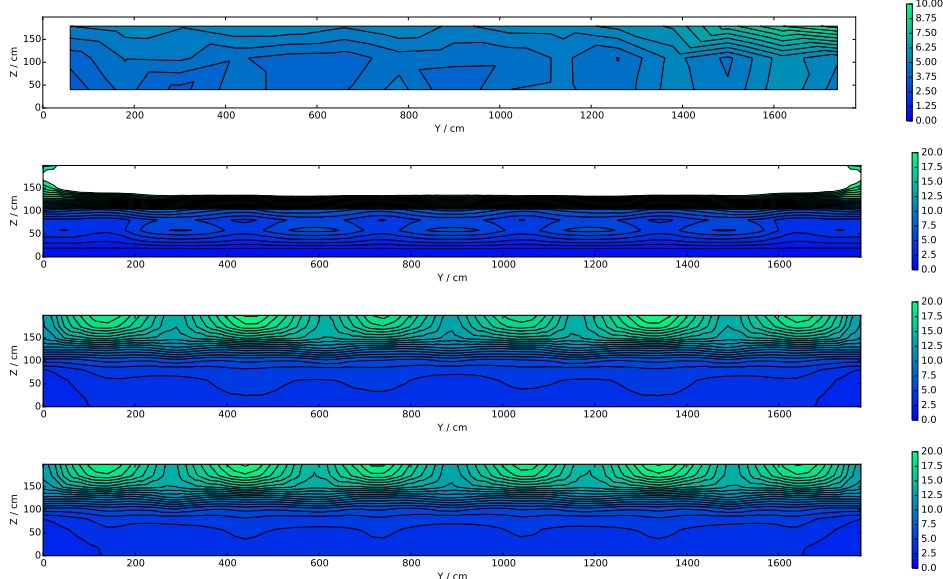

**Figure 16.** *Cont.*

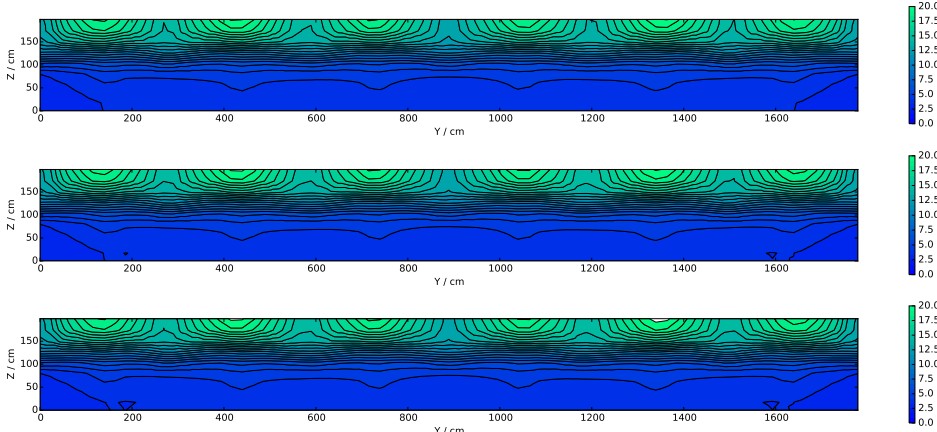

**Figure 16.** Time-averaged back wall incident radiation contours for the steady-state period of 74% HRR. Scale in $kW\,m^{-2}$. Note that for the experimental case the scale is from 0 to $10\,kW\,m^{-2}$, while for simulations it is from 0 to $20\,kW\,m^{-2}$. This change was made for visual purposes, as the M20 case does not reproduce well the experimental value of G. From **top** to **bottom**: Experimental, R4 (nPhi = nTheta = 1), R16 (nPhi = nTheta = 2), R64 (nPhi = nTheta = 4), R144 (nPhi = nTheta = 6), R256 (nPhi = nTheta = 8), and R400 (nPhi = nTheta = 10).

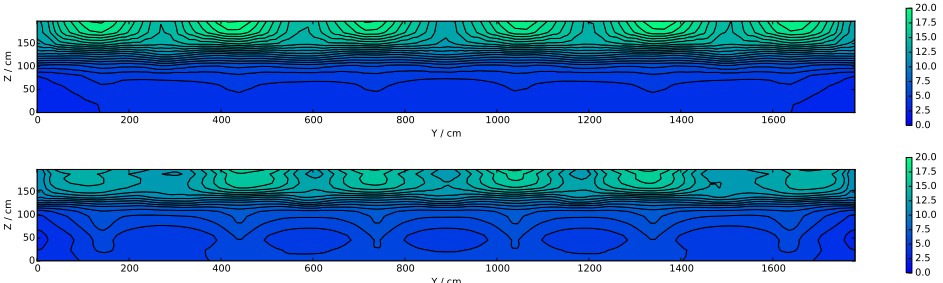

**Figure 17.** Time-averaged back wall incident radiation contours for the steady-state period of 74% HRR. **Top** to **bottom**: M20 and M10 (144 solid angles).

### 3.7. Interior Temperature Profiles

Figure 18 shows two thermocouple trees' vertical temperature profiles, including tree B2 ($x$ = 1.1 m, $y$ = 1.2 m in the near field) and tree D15 ($x$ = 2.5 m, $y$ = 9 in the far field). Regarding the central tree (D15), good agreement between the simulated and experimental data is seen between floor level and 1200 mm. At higher altitudes, fully inside the hot gas layer, temperatures are over-estimated by 75 to 150 °C with increased errors at larger heights. Some of this error can be explained by the 40 °C experimental error due to uneven propane distribution. For this far-field study, mesh convergence is achieved in the M10 case, while for the near-flame study (tree B2), no mesh convergence is observed. This is consistent, given that temperature distributions in the far field are dominated by the mixing of hot gases with quiescent air rather than chemical reactions and radiative heat transfer—a phenomenon that demands increased mesh refinement. Over-estimation of the simulated thermocouple temperatures in the hot gas zone is attributed to high values of simulated incident radiation in this sector (see Figure 16). As the *G*-field is considerably over-estimated (more than double of the experimental values), the thermocouple model delivers excessive thermocouple temperatures. The finer-meshed case M10 delivers less intensive *G*-fields near the roof (see Figure 17), and thermocouple temperature over-estimations are thus less severe.

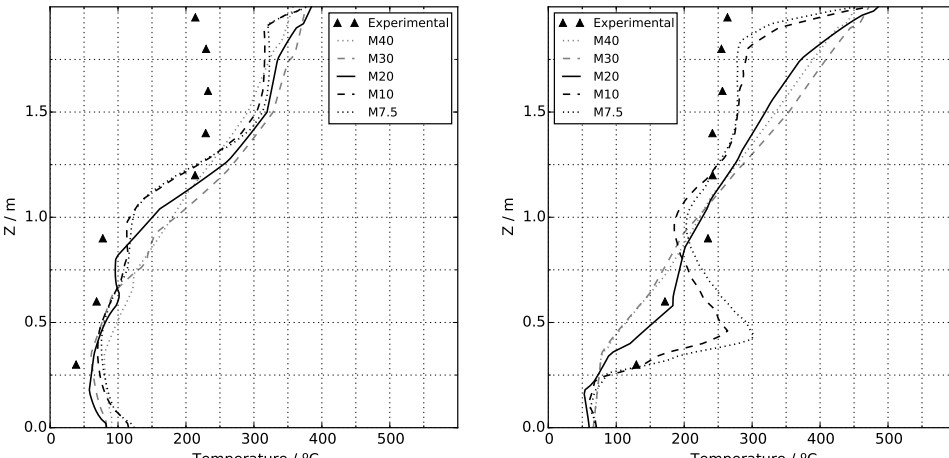

**Figure 18.** Time-averaged thermocouple temperatures for the 74% HRR steady-state period. **Left**: Central tree D15, located in the far field. **Right**: Tree B2, located along the flame-plume region of burner 6a.

As seen in the tree B2 profile, the near-flame temperature estimations are more sensitive to different mesh sizes, as in this zone combustion and temperature-induced buoyancy dominate fire dynamics [21]. Here, it can be seen how finer meshes show better agreement with experimental data, especially in the hot gas zone near the ceiling. No converged results were achieved, as refinement from M10 to M7.5 still shows considerable improvements of approximately 25 °C. Thermocouple temperature over-estimation is seen at 600 mm, with increasing error for finer meshes. Coarse meshes deliver temperatures averaged over a bigger cell than fine meshes, yielding lower reaction-zone temperatures, as coarse averaging considers zones outside this hot layer. Some of the error in the hot gas layer can be explained by the 40 °C experimental error caused by the propane flow weighting towards burner 1a, affecting burner 6a, which is adjacent to tree B2.

### 3.8. Opening Temperatures and Velocity Profiles

Results for the outflow out of the compartment are shown in Figures 19 and 20. The simulated results show global agreement to what is physically expected: at higher heat release rates, both temperatures and outflow velocities increase at the compartment outlet.

Regarding temperature profiles, it can be seen that the M10 case is mesh-converged, as refinement to M7.5 does not significantly improve the estimations. This is consistent with the observation for the central tree: mesh convergence in the far field is achieved with coarser meshes than in the flaming zone, as the dominant phenomena in the former region do not demand a high mesh resolution. Additionally, temperature estimations show less experimental agreement at the 1.18 m high thermocouple when refining the mesh. These errors increase with higher heat release rates: for M10 and M7.5 the error at 50% $HRR_{max}$ is approximately 50 °C (under-estimation), while for the 74% $HRR_{max}$ period it is approximately 75 °C (under-estimation). Even though finer meshes (cases M10 and M7.5) deliver a poor estimate of temperatures at thermocouple $z = 1.18$ m, they reproduce the steep temperature gradients at the neutral plane location better. The cases M10 and M7.5 show good agreement with fresh air entrainment temperatures—measured by the lower four thermocouples—and hence with the neutral plane height, with an over-estimation of height of less than 0.1 m. This is considering experimental and simulated neutral plane heights of 1.1 m—between 1.0 m and 1.2 m according to [23]—and 1.2 m, respectively.

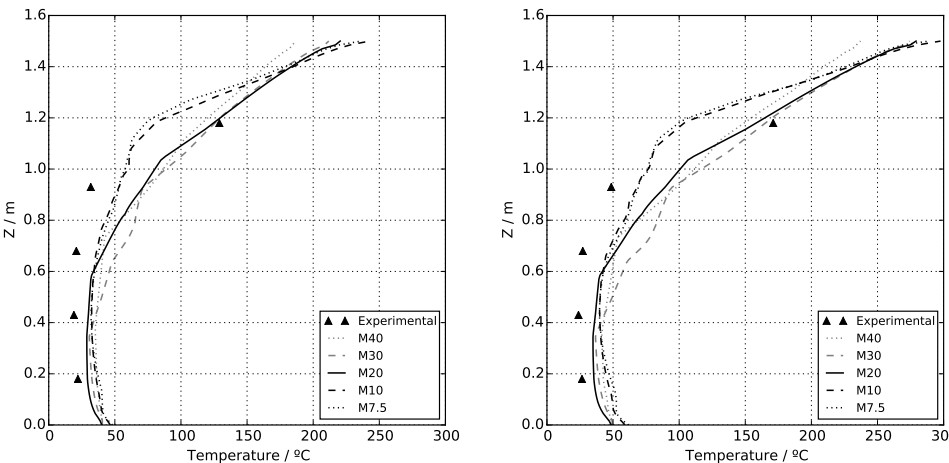

**Figure 19.** Time-averaged thermocouple temperatures at openings for the 50% (**left**) and 74% (**right**) steady-state HRR periods. Results correspond to an average between openings 1, 8, and 15.

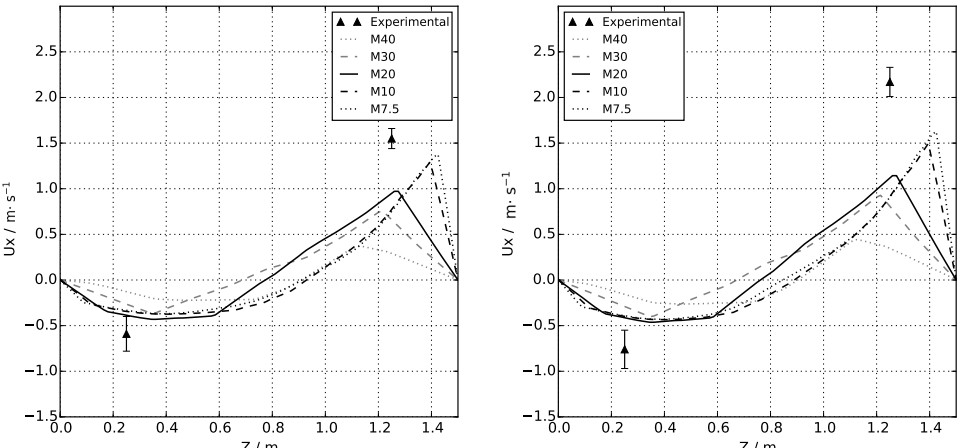

**Figure 20.** Time-averaged inflow (negative) and outflow (positve) velocities at openings for the 50% (**left**) and 74% (**right**) steady-state HRR periods. Simulation results correspond to an average between openings 1, 8, and 15. Experimental average velocities and measurement errors are taken from Hidalgo et al. [23].

A considerable under-estimation of outflow velocities is reported, especially for the coarse cases M40, M30, and M20. As simulated outflow velocities increase with the mesh size, with considerable changes even when refining from M10 to M7.5, no mesh convergence was accomplished. Regarding inflow velocities, convergence is achieved in the M30 case, with good agreement for the 50% $HRR_{max}$ scenario and slight under-estimation for the 74% $HRR_{max}$ case. The tested cases do not seem to provide enough mesh resolution to correctly capture hot gas escape dynamics, as small eddies adjacent to the hanger tip pass through the LES filter and are, therefore, modelled rather than directly simulated. This is evidenced in the fact that smaller LES filters—i.e., finer meshes—deliver better agreement with experimental data, especially near the hanger tip.

### 3.9. Temperature Evolution at Central Tree

Results for the time evolution of temperatures are shown in Figure 21. As already discussed in Figure 18, the baseline case M20 does not show good agreement with the measured thermocouple temperatures for 1.4 m high thermocouples and above, which is reflected in Figure 21. As temperature ramps in both the experimental and simulated data occur in the same time intervals, heating times associated with thermocouple thermal inertia are well captured by the transient thermocouple model. General good agreement is shown

in the 1.6 m high thermocouples and below. The temperatures measured by the 1.95 m high thermocouple do not reach steady-state conditions, an effect not captured by the simulated values which actually reach steady-state temperatures during steady-state HRR periods. As the 1.95 m high thermocouple is located within the ceiling boundary layer, its temperature is controlled by heat dissipation through the ceiling. As simulated values use the steady-state heat transfer *externalWallHeatFluxTemperature* boundary condition, which by construction cannot model the observed transient boundary layer, it is expected that the simulated boundary layer temperatures are not representative.

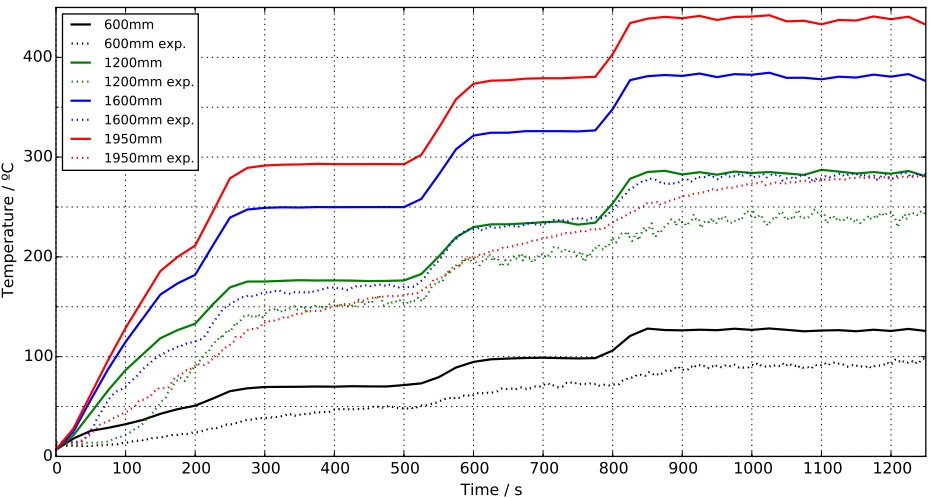

**Figure 21.** Transient thermocouple temperatures at four heights along the central tree D15. Begins at the opening of propane supply to the burners, and ends at the ending of the 99% HRR steady-state period. Case M20 compared against experimental data.

## 4. Discussion

The goal of this study is to assess FireFOAM's capabilities of simulating real-scale fires, by evaluating if the results show appropriate precision for engineering fire hazard analysis purposes. This is a necessary step in order to enable a more widespread use of FireFOAM, and ultimately take advantage of OpenFOAM's flexible meshing capabilities, including DMR. In this sense, assessing the solver's behaviour in coarse meshes that involve real-time simulations is also part of the scope of this study. In this section, the strengths and limitations of FireFOAM regarding real-scale fire modelling are discussed, also considering associated run-times. Following that, there will be a discussion on how well the simulation aligns with experimental data, taking into account the strengths and limitations discussed earlier. Additionally, relevant findings obtained from this study will be presented. Finally, the section will conclude by addressing the limitations of the methodology employed.

### 4.1. Perceived Strengths

#### 4.1.1. Real-Time Computing Capabilities

The results show that FireFOAM can run coarse meshes producing results in real time, and super real time. Among the tested cases, M30 and M40 managed to run faster than real time, where M30—consisting of 29k rectangular-prism-shaped elements in a conform mesh—simulated 750 s of fire in 676 s of clock time. As the results showed, even though both M30 and M40 were not mesh-converged and their predictions did not show good agreement with the experimental data, the flow patterns in the streamline maps and the temperature distribution in the contours showed physical consistency. These cases captured both buoyancy effects of air entrainment from the lower opening and hot gas outflow from the upper opening. Temperature maps showed temperature stratification, with increasing temperature towards the roof where the hot gas layer was located. In the far field—tested considering the central tree—both M30 and M40 showed results that diverge

by no more than ±50 °C from the finest M7.5 case. A discrepancy of this order could be simply due to temperature averaging over a bigger cell, rather than due to the coarse mesh failing to reproduce the physical phenomena. If this is the case, one can conclude that modelling the far field with coarse meshes could be perfectly acceptable, at least regarding thermodynamic properties (such as temperature).

FireFOAM's parallel processing capabilities also show promising results for reducing computing time. For the baseline M20 case with 144 solid angles in the FVDOM model, computing time was reduced from 10,000 s to 3600 s when subdividing the domain in four parts and assigning each part to a separate processor (four processors in total). A further increase to more parallel processors did not show run-time reductions, because message passing between processing units made the overall process slower when adding additional processors.

The amount of solid angles implemented in the FVDOM radiation model has an important impact on the run-time, offering another good opportunity for reduction. Considering that for ETFT test I the M20 case had a converged *G*-field with only 16 solid angles, a considerable time reduction is seen when passing from the most FVDOM demanding case (R400, 9000 s of run-time) to the converged one (R16, 3130 s). The speed-up was accomplished considering results of the same quality. It is thus vital to consider the required number of solid angles for each situation in order to obtain fast results at an acceptable level of quality.

### 4.1.2. Convergence

FireFOAM presents both solid angle and mesh convergence. As mentioned, solid angle convergence for the M20 case was achieved for 16 solid angles, considering the convergence of the *G*-field at the back wall as a sufficient criterion. Mesh convergence for internal flow fields and far-field temperatures was achieved for the M10 case. The presence of these convergence types indicates that the solver is internally consistent, given that past a certain resolution the results do not vary considerably. Plus, the simulated data tended to converge towards the experimental measurements, therefore, major errors are probably systematic— mainly due to solver input, such as gas properties, compartment thermal boundary layer properties, initial and boundary conditions, etc.—and/or due to solving relaxations, admissible error parameters, and calculation error (interpolation, differentiation and integration). A major systematic discrepancy when compared to experimental data in this study was caused by an experimental error, and is thus not attributable to deficiencies in FireFOAM.

The tested cases did not show convergence in two cases: near-flame region temperatures and compartment opening inflow/outflow velocities. In both situations, refinement from M10 to M7.5 showed, however, considerable improvement in the agreement with experimental data, therefore, finer meshes are needed in these zones of the domain to achieve good agreement. This is reasonable, as in near-flame zones fire dynamics is dominated by heat release processes and buoyant acceleration, and near the openings, velocity fields are controlled by turbulence near geometric discontinuities (e.g., near the hanger tip). These phenomena are known to require a higher mesh resolution as the LES filter has to be able to capture small eddies that control turbulent mixing, and hence reaction rates, in the near-flame zone, and to be able to capture small eddies adjacent to the hanger tip so as to correctly model the outflow dynamics. This is why mesh refinement in flame regions and in eddy-inducing geometric discontinuities is a common practice in combustion CFD. This fact highlights the advantage of dynamic mesh refinement that can be implemented in FireFOAM, where near-flame and geometric discontinuity local refinement promises to reduce computing time and deliver good-quality results.

### 4.2. FireFOAM Limitations

#### 4.2.1. Irregular-Tetrahedron Meshes

FireFOAM is relatively sensitive to the mesh topology used to discretise the domain. A converged full-time result in the T20 case (irregular mesh of 20 cm tetrahedron elements). This type of mesh construction is known to cause convergence problems in the finite

volumes discretisation concept, mainly because the method constantly evaluates fluxes through faces that are not parallel to the volume normal. This causes gradient corrections that may lead to larger numerical errors. Approximation of gradients in each element and timestep makes the solver prone to a longer computing time and divergence. This means that dynamic meshing would have to be implemented keeping the overall rectangular nature of the mesh, otherwise the time gained by the local refinement could be lost in solving complex tetrahedron geometries, or divergence could result.

### 4.2.2. Steady-State Boundary Conditions

Measured transient temperatures at 1.95 m above the floor did not achieve steady-state conditions throughout the whole extension of test I. This indicates that the ceiling jet has a characteristic gas-phase heating time higher than the time extension of any steady-state HRR periods. In order to reproduce this in simulations, a transient heat dissipation boundary condition is vital. As indicated, all cases were run using the steady-state 1D heat conduction boundary condition *externalWallHeatFluxTemperature*, which by construction cannot model the transient ceiling boundary condition observed throughout the entire extension of ETFT test I. This results in a relatively large error, given that heat dissipation through the compartment walls represents between 15 and 20% of the total energy flux leaving the compartment [24]. Implementing and validating a 1D transient diffusion equation for each patch could solve this problem, but would increase the computing time considerably as the solid-phase wall has to also be meshed and solved, and effectively a conjugate heat transfer simulation would have to be carried out. The interactions between the gas phase and solid phase is a major research area in fire science [19], but is not a problem specific to FireFOAM.

Using a heat transfer steady-state boundary condition in the transient boundary layer may also explain the energy imbalance. The problem extends to the M10 case, where in steady-state HRR periods about 10–20% of the HRR is not accounted for. Simulated heat dissipation through walls represented less than 1% of the total heat release rate, far less than the experimental 15–20% estimated by Maluk et al. [24]. This previous experimental estimation may fill the unaccounted for 10–20% gap observed in the M10 energy balance, possibly caused by the deficient modelling of the boundary layer.

### 4.2.3. Sensitivity to Domain Construction

The results were found to be sensitive to the secondary domain, i.e., the domain extension added with the purpose of minimising the influence of open boundary conditions on the in- and outflow of the compartment, and so more realistically capturing this phenomenon. It is important to note that air entrainment modelling is crucial, as differences in air entrainment into the compartment can cause considerable changes in the temperature distribution, as air contains oxidants (which control the spatial distribution of the prescribed heat release rate) and inert nitrogen (which contributes to cooling of the compartment interior).

As Figure 22 illustrates, three different secondary domain extensions were tested: one with air entrainment at compartment floor level (case A), one including the room floor with closed boundaries (case B), and one similar to the previous but with an open surface for entrainment from below the compartment (case C).

Among these cases (A, B, and C) displacements of the neutral plane were observed at the compartment opening plane, with magnitudes ranging from 0 to 10 cm, and changes in peak inflow/outflow velocities with magnitudes ranging from 0–0.2 $\mathrm{m\,s^{-1}}$. Case C showed considerable air entrainment from below the compartment floor level, and as the experimental setup contains this opening (refer to Figure 3), it was considered that this domain extension best represented the experimental conditions.

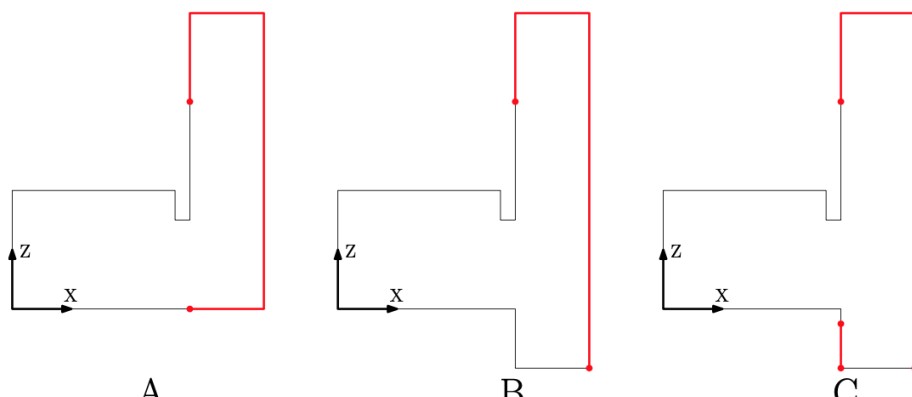

**Figure 22.** Compartment *X-Z* plane cut. Thin black lines represent wall boundaries, and thick red lines entrainment zones. Chosen cuts, for simplicity, do not include propane burners. Images not scaled to real dimensions.

Temperature and velocity profiles at the opening were both under-estimated. Inflow under-estimations were of the order of 0.1 m s$^{-1}$ (approximately 10% of the mean experimental inflow velocity) and for outflow of the order of 0.6 m s$^{-1}$ (approximately 27% of the mean experimental outflow velocity). These errors can be attributed to the fact that as air entrainment is considerably sensitive to the computational domain and to boundary conditions, the computational setup for ETFT test I has to be represented more accurately.

### 4.3. Simulated Data Agreement with Experimental Measurements

The simulated results, in general, showed reasonable agreement with the experimental data as the solver managed to capture to orders of magnitude, and in some cases even precisely reproduce, some of the measured fields. Well-reproduced phenomena included:

- Flow patterns at the compartment interior.
- Neutral plane height and pronounced temperature gradients at openings.
- Cold air/hot smoke interface height at the compartment interior.
- Temperatures in the cold air zone and in most of the hot gas zone (below the ceiling jet).

Less well reproduced phenomena were:

- Incident radiation in the hot gas zone, above the 1.4 m thermocouple.
- Temperatures in the upper half of the smoke layer, especially inside the ceiling jet.
- Transient behaviour of near-ceiling temperatures (related to the above).
- Mean inflow and outflow velocities at the compartment interior.

*G*-field over-estimations of around 100% for the M20 case are expected to be associated with coarse meshing, given the estimation improvement seen when refining to the M10 case. Simulated thermocouple temperatures in the upper half of the smoke layer were over-estimated mostly because FVDOM over-estimated the *G*-field in this zone. This is expected to originate from the under-estimation of the compartment inflow and outflow velocities. As inflow is under-estimated, less air enters the compartment to cool the interior. On the other hand, as outflow is under-estimated, hot gases do not leave the compartment at the experimental rate, and a larger accumulation of hot gases, thus, causes the ceiling radiation to increase. This is consistent, given the fact that finer meshes (M10 and M7.5) predicted higher outflow velocities, hence less intensive ceiling-level irradiation. Consequently, the simulated thermocouple temperatures were closer to the experimental measurements. Given the above, part of the error can be eliminated by mesh refinement, as finer meshes capture the flow dynamics at the opening plane more accurately. Inaccuracies in the computational domain with respect to the real experimental conditions can account for the systematic component of this error. Further systematic over-estimation of hot gas layer temperatures (even in the M7.5 and M10 cases) is attributed to the poor reproduction of heat dissipation through walls, a consequence of using the stationary 'externalWallHeat-

FluxTemperature' boundary condition where a transient boundary condition should be used. It is, however, important to note that not all discrepancies with the measurements can be attributed to the simulations. Unreported details regarding the uneven propane distribution among burners in the experiments accounted for a temperature difference of about 40 °C.

It is important to remember that when simulating real-scale fires like ETFT test I, mesh resolution is limited as large domains intrinsically demand more cells. An M5 (cell edges of 5 cm) case was planned but could not be run, as the available computing resources could not handle the almost 4 million cells that composed this mesh. Up-to-date validation exercises typically consist of smaller domains, therefore, finer meshes are viable. Vilfayeau et al. computed a $2 \times 2 \times 0.85$ m$^3$ domain with local refinement of 2–12 mm cells [18]. A finer mesh in this study could have improved results regarding near ceiling temperatures and velocity fields at openings.

### 4.4. Relevant Findings

The results reported here suggest that FireFOAM does not necessarily deliver conservative results, at least for the 20 cm and 10 cm cubic meshes. For the M10 case, mass and energy showed imbalances of +7% and 0 to −20%, respectively. M20 experienced timesteps with discontinuous solutions, therefore, it was not considered appropriate for the mass–energy balance analysis. The simulated energy distribution showed poor results, because apart from reaching −20% of unaccounted energy flows, the simulations captured almost no heat dissipation through the walls ($\leq$1% of the HRR), compared to the experimental value of 20% of the HRR [24]. The reasonably good agreement of the simulated data with the experimental data is attributed to the fact that the main energy dissipation fraction, the corresponding enthalpy of hot gases leaving the compartment, is well reproduced. The simulations indicate that on average 75% of the HRR is dissipated as the enthalpy of hot gases leaving the compartment, which is in good agreement with the experimental fraction of approximately 70% [24]. Also, supporting the previous conclusion, the simulated 8% of radiation leaving through the openings is in good agreement with the predicted experimental 10% [24]. Both of these similarities explain the well-reproduced phenomena such as flow patterns, neutral plane height, and temperatures in the cold air zone and in the lower half of the smoke layer. The temperature over-estimation in the upper half of the smoke layer—from heights 1.4 to 2 m above floor—is attributed to inaccurate simulation of flow dynamics at the compartment opening, heat dissipation through the ceiling, and unreported details of experimental problems. Inaccurate simulation of the opening plane flow dynamics due to computational domain inaccuracies yielded under-estimations of the compartment ventilation rate, hence causing over-estimations in the upper smoke-layer temperatures. Under-estimations in the ventilation rate result in less fresh air entering the compartment and less smoke evacuation, resulting in less smoke-layer cooling and elevated irradiation near the ceiling.

The results suggested that the above errors can be reduced to a certain degree by refining the numerical mesh. The main systematic errors, i.e., errors not reduced by mesh refinement, in the simulated upper smoke layer are mainly attributed to the implementation of a steady-state heat dissipation boundary condition, and to inaccuracies when reproducing experimental conditions through the computational domain (it was proven that the results are considerably sensible to its extension). The implementation of a steady-state heat dissipation model can also explain the poor energy conservation during transient periods, which is especially significant in the M20 case. This can also explain the discontinuous mass and energy fluxes obtained in transient timesteps for the M20 case. The above findings suggest that in real-scale fires, heat dissipation through walls is relevant—up to 20% of the heat release rate according to [24]—and, therefore, a one-dimensional steady-state heat transfer boundary condition is not accurate enough.

Run-time studies and results on the far field (central tree), near-flame zone (tree B2), and openings suggest that the implementation of dynamic mesh refinement in FireFOAM

could bring important benefits. Both the M30 and M40 meshes run faster than real-time (see Figure 8). On the other hand, refinement in the far field from M40 to M7.5 did not show simulation convergence towards experimental data, while in the near-flame and opening zones mesh refining implied mesh-convergence towards the experimental data with considerable improvements—up to 150 °C in the smoke layer. Considering the above, FireFOAM can reproduce the far field reasonably well using real-time running meshes, and at the same time could benefit from refinement in the near-flame zone and in the vicinity of geometric discontinuities, such as the hanger tip. As this methodology is the principle of dynamic mesh refinement, the results presented here support the idea of implementing it in FireFOAM in scenarios where the fire front moves, and dynamic mesh refinement could be beneficial.

## 5. Conclusions

A detailed validation of FireFOAM as a tool to simulate real-scale fuel-controlled fires in large open-plan compartments, complementing considerations of accuracy with an assessment of the solver performance in terms of run-time under different solver conditions is presented. These conditions mainly consisted of different meshes—different in size and element geometry, different FVDOM resolutions—controlled through the number of solid angles, and different numbers of parallel processors for computing results. Both theory-based and experiment-based techniques were implemented. Theory-based validation included mass balances, energy balances, and streamline maps to test the internal consistency of FireFOAM. Experiment-based validation consisted of a comparison of the simulation results with the ETFT test I measurements [23], including variables such as temperature, incident radiation, and velocity fields. It was found that a 10 cm rectangular prism mesh with 144 solid angles in FVDOM is sufficient to accurately capture the main phenomena of a full-scale fuel-controlled compartment fire.

FireFOAM demonstrated adequate real-time computational and convergence capabilities, though it is primarily constrained by mesh characteristics (such as geometry and element orientation) and presents a high sensitivity to the domain setup and boundary conditions. The simulated results showed reasonable agreement with experimental data, capturing magnitudes accurately and in some cases even replicating measured fields to a significant precision. Furthermore, the study pointed out limitations of the solver and its application, emphasising the factors contributing to the discrepancies observed between the simulated results and experimental data.

*Future Work*

Speeding up fire simulations is important for many aspects of fire safety science. In this work, FireFOAM has shown good potential for reducing run-time through parallel processing and mesh refinement. This opens the door for future work on implementing DNR in FireFOAM when simulating fire scenarios with flame spread/extinction, e.g., for fire scenario forecasting or simulating travelling fires [35]. Several strengths and constraints of FireFOAM were identified, providing guidelines for future investigators in this line of work. Recommendations for future work include:

- A revision of the effects of subgrid-scale models (e.g., one-equation eddy viscosity model) on energy conservation, as in these exercises the finite volumes method was not found to be conservative, with errors ranging from 10–20%.
- Investigate the implementation of conjugate heat transfer simulations to better account for heat losses, and consequently improve near wall temperature estimations.
- Refine using mesh resolutions in the 1–15 mm range, as is common practice in other CFD applications. This is to confirm complete mesh convergence. In this study, it was seen how the M7.5 case (75 mm) was not mesh convergent for near-flame zones and in the vicinity of geometric discontinuities.

- Following the previous recommendation, implement mesh refinement in the near-flame zone and near geometric discontinuities, mesh coarsening in the far-field and flame-extinction zones, and assess speed-up and improvements in overall performance.

Finally, as meshing times are negligible compared to computing times, DMR proves to be an excellent opportunity for reducing computing times while obtaining good-quality results. As OpenFOAM uses a solver capable of computing irregular-meshed domains and subdividing these domains for parallel processing, future work in this line of investigation has to take advantage of these two capabilities. Most commercial solvers do not provide this level of flexibility.

**Author Contributions:** Conceptualization, W.J. and R.Z.; methodology, W.J. and R.Z.; software, R.Z.; validation, R.Z. and W.J.; formal analysis, R.Z. and W.J.; investigation, R.Z., I.C., R.C. and B.M.; resources, W.J.; data curation, R.Z.; writing—original draft preparation, R.Z., W.J., I.C., R.C. and B.M.; writing—review and editing, R.Z., W.J., I.C., R.C. and B.M.; visualisation, R.Z. and W.J.; supervision, W.J.; project administration, W.J.; funding acquisition, W.J. All authors have read and agreed to the published version of the manuscript.

**Funding:** The APC was funded by ANID BASAL FB210015 (CENAMAD).

**Institutional Review Board Statement:** Not applicable.

**Informed Consent Statement:** Not applicable.

**Data Availability Statement:** No new data were created.

**Acknowledgments:** W. Jahn and I. Calderón would like to recognise the financial support provided by the Timber Innovation Center UC and ANID BASAL FB210015 (CENAMAD). We also thank The University of Edinburgh for making the data of the ETFT available for everyone to use.

**Conflicts of Interest:** The authors declare no conflict of interest.

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
