# Peer review of "Assessment of the Performance of FireFOAM in Simulating a Real-Scale Fire Scenario Using High Resolution Data"

_fire, doi:10.3390/fire6100375_

Round 1
Reviewer 1 Report
Very interesting material. Well developed.
Author Response
Dear reviewer,
Thanks you for reviewing our manuscript and for your encouragement.
Reviewer 2 Report
The reviewed article presents a well-elaborated study, which assesses the performance of fireFOAM in simulating a large-scale fire scenario, namely using the Edinburgh Tall Building Fire Test. The manuscript meets the required criteria of a scientific research article suitable for publication in the given professional journal. The “Abstract” correctly summarizes content of the article. The chapter “Introduction” familiarizes the reader with the basic theoretical starting points of the solved problem. The second chapter "Material and Methods" in several subsections describes in detail the applied numerical solver, obtained experimental data, numerical setup, and data comparison. The authors' own contribution is contained in the third chapter, which, within its nine sub-chapters, provides a detailed description and analysis of the obtained results. The fourth chapter provides a broad discussion of the achieved results. The article is elaborated on an extensive area of almost 30 pages. The text is appropriately supplemented by 22 illustrative figures. I consider the list of 24 references to be sufficient. It is evident, with regard to overall processing of the article, its methodology, organization and presentation of the obtained research results, that the authors are professionally competent and well-oriented in the given professional area. I have no relevant comments or suggestions for improvement of the article. I recommend publishing it as presented, without changes or supplements.
Author Response
Dear Reviewer,
Thanks you for reviewing our manuscript and for your encouragement.
Reviewer 3 Report
My comments are as follows, after reviewing the manuscripts:
1. I could not find the significance of the study in this manuscript, the authors developed a model of a fire which can be modelled by other software and noticed some limitations, so how the readers will benefit from this manuscript if other software are able to model the fire.
2. at page 3 line number 91 it is mentioned "Low Mach Number compressible Navier-Stokes equations" could you please explain what you mean by low Mach & compressible?
3. validation with experiments only shown for temperature, how about heat flux validation and species & gases generated from fire validation, why it was not shown?
4. How can you prove that your OpenFOAM model is able to model ventilation controlled fire?
5. Figure 14 shows the streamlines which appears to be qualitative validation, how about quantitative validation against measurements?
English is acceptable.
Author Response
Dear Reviewer 3: We thank you for your valuable comments, addressing them helped us to greatly improve the manuscript. Below are the replies to each of your questions/comments, specifying also where the manuscript was adjusted (all added changes will be in blue in the revised manuscript). 1. I could not find the significance of the study in this manuscript, the authors developed a model of a fire which can be modelled by other software and noticed some limitations, so how the readers will benefit from this manuscript if other software are able to model the fire. After reading the manuscript again we acknowledge that this was indeed not as clear as intended. Thank you for pointing it out. The main goal of the paper is to provide validation of FireFOAM by comparing the results to real-scale, heavily instrumented fire tests, which enable comparison not only at an integrated level, but also in detail (thermocouple racks at different locations inside the compartment, or velocity profiles at openings). Additionally we wanted to analyse the performance of FireFOAM, and correlate its mesh dependence. A final goal of the study was to assess the sensitivity to various input parameters related to the numerical set-up. The initial motivation for this was not only the enabling of an alternative software per se (there is no real need to replace FDS in most applications), but to provide an alternative for cases where, by construction, FDS touches boundaries. This is the case with flexible meshing, which in our opinion is a promising feature for speeding up fire simulations. The manuscript was adjusted and text was added to clarify all of the above. 2. at page 3 line number 91 it is mentioned "Low Mach Number compressible Navier-Stokes equations" could you please explain what you mean by low Mach & compressible? This is a very important question, we agree that discussing this point is crucial. We have added a paragraph to the manuscript to explain this to more detail. Usually, low Mach number solver do not couple density changes with instantaneous pressure changes, i.e. they solve for incompressible flow. To be able to simulate thermally driven buoyant flows with incompressible solvers, the Boussinesq approximation is invoked. This approximation, however, is only valid for small temperature variations. Density variations due to heat addition, for instance as a result of chemical reactions are generally much larger, and a less restrictive approach is needed. One such approach is to apply the equations of motion propoused by Rehm & Baum, considering a spatially uniform mean pressure appearing in both energy equation and the equation of state, and the spatially non-uniform portion of the pressure appearing in the momentum equation. Therefore, the pressure remains almost constant in space while significant density and temperature variations are allowed (Rehm, 1978). Initially, this algorithm was designed to study buoyant plumes at the Boussinesq limit, meaning the fluid is considered incompressible, but a source term is included to account for buoyancy effects. Eventually, this approach proved to be too limiting, but some of the key features of the algorithm were retained, such as the low Mach number (FDS Technical Reference Guide Sixth Edition, 2023). The approach implemented in OpenFOAM is similar in nature (Hassanaly, Koo, Lietz, Chong, & Raman, 2018). 3. validation with experiments only shown for temperature, how about heat flux validation and species & gases generated from fire validation, why it was not shown? We do not agree with this comment, as much more than temperature comparison is done and shown in the paper. Figures 16 and 17 of the paper actually show incident heat fluxes at the back wall of the compartment, and the results are discussed in the text. Also, in figure 20 time-averaged inflow and outflow velocities at the openings are shown and compared. All these instances of comparison are explained in the section "results" and their respective subsection. There was not data for species generation to validate against. 4. How can you prove that your OpenFOAM model is able to model ventilation controlled fire? The experiments used in this study to validate FireFOAM simulations were largely fuel controlled, not ventilation controlled. In that sense we do not really know how to answer this particular question. Ventilation controlled fires, or really regime I type fires, are indeed much more difficult to reproduce with CFD based fire models, due to the insufficient numerical resolution that affects flame extinction. It is, however, important to note that it is arguably much more interesting to simulate ventilation controlled fires since in large open-plan spaces more spatial variation in the variables of interest is to be expected. As mentioned above, one of the goals of this study was to enable the use of a solver with higher meshing flexibility for fire simulations, as this spatial variation could allow for less restolution in certain zones of the domain. 5. Figure 14 shows the streamlines which appears to be qualitative validation, how about quantitative validation against measurements? This is actually something we would have very much liked to do. Unfortunately we did not have data to validate against. In the experiments, velocity probes were installed at the openings only, therefore infering streamlines inside the compartment from data was not possible. For this reason, we only included a qualitative analysis. We added text to clarify this point in the subsection 3.4 between lines 408 and 410.Round 2
Reviewer 3 Report
The authors have made the required corrections and properly responded to the comments.